# RGFN: Synthesizable Molecular Generation Using GFlowNets

**Michał Koziarski**[*,1,2], **Andrei Rekesh**[*,3], **Dmytro Shevchuk**[*,3], **Almer van der Sloot**[1,2],
**Piotr Gaiński**[4,1,2], **Yoshua Bengio**[1,2], **Cheng-Hao Liu**[1,5], **Mike Tyers**[6,3], **Robert A. Batey**[3,7]

[1] Mila – Québec AI Institute, [2] Université de Montréal, [3] University of Toronto,
[4] Jagiellonian University, [5] McGill University, [6] The Hospital for Sick Children Research Institute,
[7] Acceleration Consortium, [*] Equal contribution

`michal.koziarski@mila.quebec`, `{a.rekesh,dmytro.shevchuk}@mail.utoronto.ca`

## Abstract

Generative models hold great promise for small molecule discovery, significantly increasing the size of search space compared to traditional in silico screening libraries. However, most existing machine learning methods for small molecule generation suffer from poor synthesizability of candidate compounds, making experimental validation difficult. In this paper we propose Reaction-GFlowNet (RGFN), an extension of the GFlowNet framework that operates directly in the space of chemical reactions, thereby allowing out-of-the-box synthesizability while maintaining comparable quality of generated candidates. We demonstrate that with the proposed set of reactions and building blocks, it is possible to obtain a search space of molecules orders of magnitude larger than existing screening libraries coupled with low cost of synthesis. We also show that the approach scales to very large fragment libraries, further increasing the number of potential molecules. We demonstrate the effectiveness of the proposed approach across a range of oracle models, including pretrained proxy models and GPU-accelerated docking.

## 1 Introduction

Traditionally, machine learning has been applied to drug discovery for screening existing libraries of compounds, whether actual physical collections or pre-configured in silico collections of readily synthesizable compounds, in a supervised fashion. However, supervised screening of the whole drug-like space, often estimated to contain approximately $10^{60}$ [41] different compounds, is infeasible in practice. Generative methods offer the potential to circumvent this issue by sampling directly from a distribution over desirable chemical properties without the need to evaluate every possible molecular structure. Despite these advances, existing generative approaches tend not to explicitly enforce synthesizability [21], generating samples that might be either very costly or altogether impossible to chemically synthesize. Ensuring that generative methods operate in the space of synthesizable compounds, yet at a much larger and more diverse scale than existing chemical libraries, remains an open challenge.

In this paper, we propose Reaction-GFlowNet (RGFN), an extension of the GFlowNet framework [5] that generates molecules by combining basic chemical fragments using a chain of reactions. We propose a relatively small collection of cheap and accessible chemical building blocks (reactants), as well as established high-yield chemical transformations, that together can still produce a search space orders of magnitude larger than existing chemical libraries. We additionally propose several domain-specific extensions of the GFlowNet framework for action representation and scaling to a larger space of possible actions.

Source code available at `https://github.com/koziarskilab/RGFN`.

38th Conference on Neural Information Processing Systems (NeurIPS 2024).

We experimentally evaluate RGFN on a set of diverse screening tasks, including docking score approximation with a trained proxy model for soluble epoxide hydrolase (sEH), GPU-accelerated direct docking score calculations for multiple protein targets (Mpro, ClpP, TBLR1 and sEH), and biological activity estimation with a trained proxy model for dopamine receptor type 2 (DRD2) receptor activity and senolytic activity [74]. We demonstrate that RGFN produces similar optimization quality and diversity to existing fragment-based approaches while ensuring straightforward synthetic routes for predicted hit compounds.

## 2 Related work

**Generative models for molecular discovery.** A plethora of methods have been developed for molecular generation [48, 7] using machine learning. These methods can be categorized depending on the molecular representation used, including textual representations such as SMILES [34, 3, 39], molecular graphs [33, 46, 54] or 3D atom coordinate representations [52], as well as the underlying methodology, for example, variational autoencoders [33, 46], reinforcement learning [54, 37] or diffusion models [60]. Recently, Generative Flow Networks (GFlowNets) [4, 50, 59, 64, 72, 19] have emerged as a promising paradigm for molecular generation due to their ability to sample large and diverse candidate small molecule space, which is crucial in the drug discovery process. Traditionally, GFlowNets for molecular generation operated on the graph representation level, and candidate molecules were generated as a sequence of actions in which either individual atoms or small molecular fragments were combined to form a final molecule. While using graph representations, as opposed to textual or 3D representations, allows the enforcement of the validity of the generated molecules, it does not guarantee a straightforward route for chemical synthesis. Here, we expand on the GFlowNet framework by modifying the space of actions to consist of choosing molecular fragments and executing compatible chemical reactions/transformations, in turn guaranteeing both physical-chemical validity and synthesizability.

**Synthesizability in generative models.** One approach to ensuring the synthesizability of generated molecules is by using a scoring function, either utilizing it as one of the optimization criteria [37], or as a post-processing step for filtering generated molecules. Multiple scoring approaches, both heuristic [18, 23] and ML-based [42], exist in the literature. Another branch of research focuses on using reaction models and traversing predicted synthesis graph [8, 12, 38, 55]. However, reaction and synthesizability estimation is difficult in practice, and can fail to generalize out-of-distribution in the case of ML models. Furthermore, theoretical synthesizability does not necessarily account for the cost of synthesis. Because of this, a preferable approach might be to constrain the space of possible molecules to those easily synthesized by operating in a predefined space of chemical reactions and fragments. Several recent strategies employ this approach [22, 49, 67], including reinforcement learning-based methods [24, 27] and a concurrent work utilizing GFlowNets [15]. We extend this line of investigation not only by translating the concept to the GFlowNet framework but also by proposing a curated set of robust chemical reactions and fragments that ensure efficient synthesis at low total costs.

## 3 Method

### 3.1 Generative Flow Networks

GFlowNets are amortized variational inference algorithms that are trained to sample from an unnormalized target distribution over compositional objects. GFlowNets aim to sample objects from a set of terminal states $\mathcal{X}$ proportionally to a reward function $\mathcal{R} : X \to \mathbb{R}^+$. GFlowNets are defined on a pointed directed acyclic graph (*DAG*), $G = (S, A)$, where:

- $s \in S$ are the nodes, referred to as states in our setting, with the special starting state $s_0$ being the only state with no incoming edges, and the terminal states $\mathcal{X}$ have no outgoing edges,
- $a = s \to s' \in A$ are the edges, referred to as actions in our setting, and correspond to applying an action while in a state $s$ and landing in state $s'$.

We can define a non-negative flow function on the edges $F(s \to s')$ and on the states $F(s)$ of the DAG such that $\forall x \in \mathcal{X} F(x) = \mathcal{R}(x)$. A perfectly trained GFlowNet should satisfy the following flow-matching constraint:

$$\forall s \in S \quad F(s) = \sum_{(s'' \to s) \in A} F(s'' \to s) = \sum_{(s \to s') \in A} F(s \to s'). \tag{1}$$

A state sequence $\tau = (s_0 \to s_1 \to \ldots \to s_n = x)$, with $s_n = x \in \mathcal{X}$ and $a_i = (s_i \to s_{i+1}) \in A$ for all $i$, is called a complete trajectory. We denote the set of trajectories as $\mathcal{T}$.

Another way to rephrase the flow-matching constraints is to learn a forward policy $P_F(s_{i+1}|s_i)$ such that trajectories starting at $s_0$ and taking actions sampled by $P_F$ terminate at $x \in \mathcal{X}$ proportional to the reward.

**Trajectory balance.** Several training losses have been explored to train GFlowNets. Among these, trajectory balance [45] has been shown to improve credit assignment. In addition to learning a forward policy $P_F$, we also learn a backward policy $P_B$ and a scalar $Z_\theta$, such that, for every trajectory $\tau = (s_0 \to s_1 \to \ldots \to s_n = x)$, they satisfy:

$$Z_\theta \prod_{t=1}^{n} P_F(s_t|s_{t-1}) = R(x) \prod_{t=1}^{n} P_B(s_{t-1}|s_t) \tag{2}$$

## 3.2 Reaction-GFlowNet

Reaction-GFlowNet generates molecules by combining basic chemical fragments using a chain of reactions. The generation process comprises the following steps (illustrated in Figure 1):

1. Select an initial building block (reactant or surrogate reactant; see Appendix C.2 for more details about surrogate reactants).
2. Select the reaction template (a graph transformation describing the reaction).
3. Select another reactant.
4. Perform the in silico reaction and select one of the resulting molecules.
5. Repeat steps 2-4 until the stop action is selected.

In the rest of this section, we describe the design of the Reaction-GFlowNet in detail.

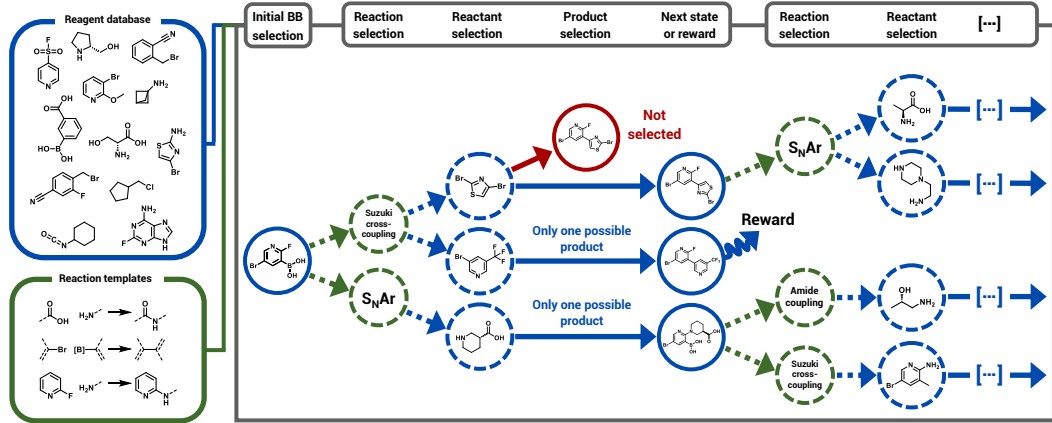

Figure 1: Illustration of RGFN sampling process. At the beginning, the RGFN selects an initial molecular building block. In the next two steps, a reaction and a proper reactant are chosen. Then the in silico reaction is simulated with RDKit's RunReactants functionality and one of the resulting molecules is selected. The process is repeated until the stop action is chosen. The obtained molecule is then evaluated using the reward function.

**Preliminaries.** Reaction-GFlowNet uses a predefined set of reaction patterns and molecules introduced in Section 3.3. We denote these as $R$ and $M$ respectively. As a backbone for our forward policy $P_F$, we use a graph transformer model $f$ from [77]. The graph transformer takes as an input a molecular graph $m$ and outputs the embedding $f(m) \in \mathbb{R}^D$, where $D$ is the embedding dimension.

In particular, $f$ can embed an empty graph $\emptyset$. It can additionally be conditioned on the reaction $r \in R$ which we denote as $f(m, r)$. The reaction in this context is represented as its index in the reaction set $R$.

**Select an initial building block.** At the beginning of each trajectory, Reaction-GFlowNet selects an initial fragment from the collection of building blocks $M$. The probability of choosing $i$-th fragment $m_i$ is equal to:

$$p(m_i|\emptyset) = \sigma^{|M|}(\mathbf{s})_i, \ s_i = \text{MLP}_M(f(\emptyset))_i, \tag{3}$$

where $\text{MLP}_M : \mathbb{R}^D \to \mathbb{R}^{|M|}$ is a multi-layer perceptron (MLP). The $\sigma^k$ is a standard softmax over the logits vector $\mathbf{s} \in \mathbb{R}^k$ of the length $k$:

$$\sigma^k(\mathbf{s})_i = \frac{\exp(s_i)}{\sum_{j=1}^k \exp(s_j)}.$$

**Select the reaction template.** The next step is to select a reaction that can be applied to the molecule $m$. The probability of choosing $i$-th reaction from $R$ is described as:

$$p(r_i|m) = \sigma^{|R|+1}(\mathbf{s})_i, \ s_i = \text{MLP}_R(f(m))_i, \tag{4}$$

where $\text{MLP}_R : \mathbb{R}^D \to \mathbb{R}^{|R|+1}$ is an MLP that outputs logits for reactions from $R$ and an additional stop action with index $|R| + 1$. Choosing the stop action in this phase ends the generation process. Note that not all the reactions may be applied to the molecule $m$. We appropriately filter such reactions and assume that the score $s_i$ for non-feasible reactions is equal to $-\infty$.

**Select another reactant.** We want to find a molecule $m_i \in M$ that will react with $m$ in the reaction $r$. The probability for selecting $m_i$ is defined as:

$$p(m_i|m, r) = \sigma^{|M|}(\mathbf{s})_i, \ s_i = \text{MLP}_M(f(m, r))_i \tag{5}$$

where $\text{MLP}_M$ is shared with the initial fragment selection phase. As in the previous phase, not all the fragments can be used with the reaction $r$, so we filter these out.

**Perform the reaction and select one of the resulting molecules.** In this step, we apply the reaction $r$ to the two fragment molecules chosen in previous steps. As the reaction pattern can be matched to multiple parts of the molecules, the result of this operation is a set of possible outcomes $M'$. We choose the molecule $m_i' \in M'$ by sampling from the following distribution:

$$p(m_i') = \sigma^{|M'|}(\mathbf{s})_i, \ s_i = \text{MLP}_{M'}(f(m_i')), \tag{6}$$

where $\text{MLP}_{M'} : \mathbb{R}^D \to \mathbb{R}$ scores the embedded $m_i'$ molecule.

**Backward Policy.** A backward policy in RGFN is only non-deterministic in states corresponding to a molecule $m$ which is a result of performing some reaction $r \in R$ on molecule $m'$ and reactant $m'' \in R$. We denote the set of such tuples $(r, m', m'')$ that may result in $m$ as $T$. We override the indexing and let $(r_i, m_i', m_i'')$ be the $i$-th tuple from $T$. The probability of choosing the $i$-th tuple is:

$$p((r_i, m_i', m_i'')|m) = \sigma^{|T|}(\mathbf{s})_i, \ s_i = \text{MLP}_B(f(m_i', r_i)), \tag{7}$$

where $\text{MLP}_B : \mathbb{R}^D \to \mathbb{R}$ and $f$ is a backbone transformer model similar to the one used in the forward policy. To properly define $T$, we need to implicitly keep track of the number of reactions performed to obtain $m$ (denoted as $k$). Only those tuples $(r, m', m'')$ are contained in the $T$ for which we can recursively obtain $m'$ in $k - 1$ reactions.

**Action Embedding.** While the $\text{MLP}_M$ used to predict the probabilities of selecting a molecule $m_i \in M$ works well for our predefined $M$, it underperforms when the size of possible chemical building block library is increased. Such an $\text{MLP}_M$ likely struggles to reconstruct the relationship between the molecules. Intuitively, when a molecule $m_i$ is chosen in some trajectory, the training signal from the loss function should also influence the probability of choosing a structurally similar $m_j$. However, the $\text{MLP}_M$ disregards the structural similarity by construction and it intertwines the probabilities of choosing $m_i$ and $m_j$ only with the softmax function. To incorporate the relationship between molecules into the model, we embed the molecular building blocks with a simple machine learning model $g$ and reformulate the probability of choosing a particular building block $m_i$:

$$p(m_i|m, r) = \sigma^{|M|}(\mathbf{s})_i, \ s_i = \phi(Wf(m, r))^T g(m_i), \tag{8}$$

where $\phi$ is some activation function (we use GELU) and $W \in \mathbb{R}^{D \times D}$ is a learnable linear layer. Note that if we define $g(m_i)$ as an index embedding function that simply returns a distinct embedding for every $m_i$, we will obtain a formulation equivalent to Equation (5). To leverage the structure of molecules during the training, we use $g$ that linearly embeds a MACSS fingerprint [40] of an input molecule $m_i$ along with the index $i$. Note that this approach does not add any additional computational costs during the inference as the embeddings $g(m_i)$ can be cached. In Section 4.3, we show that this method greatly improves the performance when scaling to larger sets of fragments.

### 3.3 Chemical language

We select seventeen reactions and 350 building blocks for our model. These include amide bond formation, nucleophilic aromatic substitution, Michael addition, isocyanate-based urea synthesis, sulfur fluoride exchange (SuFEx), sulfonyl chloride substitution, alkyne-azide and nitrile-azide cycloadditions, esterification reactions, urea synthesis using carbonyl surrogates, Suzuki-Miyaura, Buchwald-Hartwig, and Sonogashira cross-couplings, amide reduction, and peptide terminal thiourea cyclization reactions to produce iminohydantoins and tetrazoles. The chosen reactions are known to be typically quite robust [9] and are often high-yielding (75-100%), thus enforcing reliable synthesis pathways when sampling molecules from our model. To simulate couplings in Python, reactions are encoded as SMARTS templates. To ensure compatible building blocks yielding specific, chemically valid products and to enable parent state computation, we introduce multiple variants corresponding to differing reagent types for most of the proposed reactions. In some cases SMARTS templates encode reactions where one of the reagents is not specified. We describe these transformations as *implicit reactions* (Appendix E). Additionally, we introduce *surrogate reactions* where the SMARTS templates and SMILES strings encode for alternate building blocks (see Appendix C.2 for more details). Finally, once again for the sake of specificity, reactions are duplicated by swapping the order of the building block reactants. In total, 132 different SMARTS templates are used.

During the construction of the curated building block database, only affordable reagents (i.e., building blocks) are considered. For the purposes of this study we define affordable reagents to be those priced at less than or equal to $200 per gram. The mean cost per gram of reagents selected for this study is $22.52, the lowest cost $0.023 per gram, and the highest cost $190 per gram (see Appendix N for more details on cost estimation).

A crucial consideration when choosing the set of reactions and fragments used is the state space size (the number of possible molecules that can be generated using our framework). This is difficult to compute precisely since a different set of reactions or building blocks is valid for every state in a given trajectory. We estimate this based on 1,000 random trajectories instead (details can be found in Appendix A). In addition to our 350 low-cost fragments, we also perform this analysis with 8,000 additional random Enamine building blocks. Comparison for different numbers of maximum reactions is presented in Figure 2. We demonstrate that even with curated low-cost reactants and limiting the number to a maximum of four reactions, state space size is an order of magnitude greater than the number of molecules contained in Enamine REAL [17]. This size can increase significantly with the addition of more fragments and/or an increase in the maximum number of reactions. Additional discussions regarding scaling can be found in Section 4.3.

## 4 Experimental study

In the conducted experiments we compare oracle scores and synthesizability scores of RGFN with several state-of-the-art reference methods. Secondly, we examine the capabilities of RGFN to scale to larger fragment libraries, in particular when using the proposed action embedding mechanism. Finally, we perform an in-depth examination of generated ligands across several biologically relevant targets.

### 4.1 Set-up

Throughout the course of the conducted computational experimental study, we aim to evaluate the performance of the proposed approach across several diverse biological oracles of interest. This includes proxy models (machine learning oracles, pretrained on the existing data and used for higher computational efficiency): first, the commonly used sEH proxy as described in [4]. Second, a graph

neural network trained on the biological activity classification task of senolytic [74] recognition. Third, the Dopamine Receptor D2 (DRD2) oracle [53] from Therapeutics Data Commons [28]. Proxy model details are provided in Appendix B.1.

Per the GFlowNet training algorithm, the reward is calculated for a batch of dozens to hundreds of molecules at each training step, rendering traditional computational docking score algorithms like AutoDock Vina [70] infeasible for very large training runs. As a result, previous applications of GFlowNets to biological design [4, 64] employed a fast pre-trained proxy model trained on docking scores instead. These proxies, while lightweight, present potential issues should the GFlowNet generate molecules outside their training data distributions and require receptor-specific datasets. To circumvent this, we use the GPU-accelerated Vina-GPU 2.1 [68] implementation of the QuickVina 2 [2] docking algorithm to calculate docking scores directly in the training loop of RGFN. This approach allows for drastically increased flexibility in protein target selection while eliminating proxy generalization failure. We selected X-ray crystal structures of human soluble epoxy hydrolase (sEH), ATP-dependent Clp protease proteolytic subunit (ClpP), SARS-CoV-2 main protease (Mpro), and transducin $\beta$-like-related protein 1 as targets for

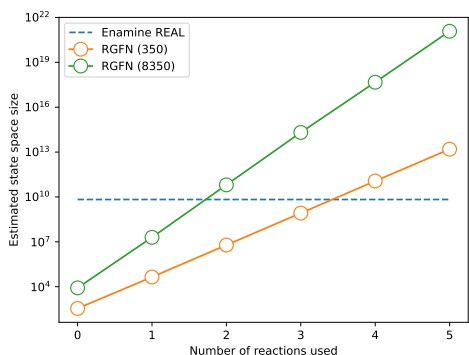

Figure 2: Estimation of the state space size of RGFN as a function of the maximum number of allowed reactions. RGFN (350) indicates a variant using 350 hand-picked inexpensive building blocks, while RGFN (8350) also uses 8,000 randomly selected Enamine building blocks. Enamine REAL (6.5B compounds) is shown as a reference.

evaluating RGFN using a docking reward (detailed motivation for specific target selection is provided in Appendix G).

## 4.2 Comparison with existing methods

We begin experimental evaluation with a comparison to several state-of-the-art methods for molecular discovery. Specifically, we consider a genetic algorithm operating on molecular graphs (GraphGA) [32] as implemented in [10], which has been demonstrated to be a very strong baseline for molecular discovery [21], Monte Carlo tree search-based SyntheMol [67], cascade variational autoencoder (casVAE) [49], and a fragment-based GFlowNet (FGFN) [4] as implemented in [57]. For FGFN, we additionally considered its variant that had a SAScore as one of the reward terms (FGFN+SA). Training details can be found in Appendix B.2. It is worth noting that besides SyntheMol, which also operates in the space of chemical reactions and building blocks derived from the Enamine database, and casVAE, which used the set of reaction trees obtained from the USPTO database [43], our remaining benchmarks do not explicitly enforce synthesizability when generating molecules. Because of this, in this section, we will examine not only the quality of generated molecules in terms of optimized properties but also their synthesizability. We consider only two reaction-based approaches, as other existing methods employing this paradigm [27, 24] do not share code or curated reactions and building blocks, making reproduction difficult.

We first examine the distributions of rewards found by each method across four different oracles used for training: sEH proxy, senolytic proxy, DRD2 proxy, and GPU-accelerated docking for ClpP. The results are presented in Figure 3. As can be seen, while RGFN underperforms in terms of average reward when compared to the method not enforcing synthesizability (GraphGA), it outperforms SyntheMol's and casVAE's reaction-based sampling. Interestingly, when compared to standard FGFN, RGFN either performs similarly (ClpP docking) or achieves higher average rewards. This is most striking in the case of the challenging senolytic discovery task, in which a proxy is trained on a severely imbalanced dataset with less than 100 actives, resulting in a sparse reward function. We suspect that this, possibly combined with a lack of compatibility between the FGFN fragments and known senolytics, led to the failure to discover any high-reward molecules. However, RGFN succeeds in the task and finds a wide range of senolytic candidates. Finally, the gap in performance is

even larger between RGFN and FGFN+SA, indicating that introducing synthesizability constraints reduces the ability of FGFN to discover high-reward molecules.

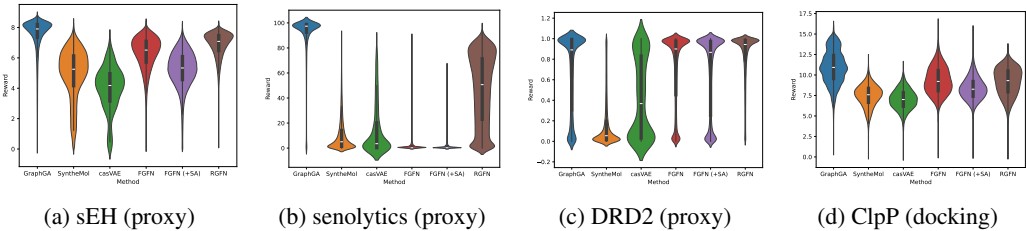

| (a) sEH (proxy) | (b) senolytics (proxy) | (c) DRD2 (proxy) | (d) ClpP (docking) |

Figure 3: Distributions of rewards across different tasks.

Secondly, we examine the number of discovered modes for each method, with a mode defined as a molecule with computed reward above a threshold (sEH: 7, senolytics: 50, DRD2: 0.95, ClpP docking: 10), and Tanimoto similarity to every other mode $< 0.5$. We use Leader algorithm for mode computation. The number of discovered modes across tasks as a function of normalized iterations is presented in Figure 4. Note that in the case of GraphGA, FGFN, FGFN+SA, and RGFN this simply translates to the number of oracle calls, but for SyntheMol and casVAE, due to large computational overhead, we impose a maximum number of oracle calls such that training time was comparable to RGFN (see Appendix B.2 for details). Note that in the case of casVAE, this resulted in a very small number of molecules being visited in the allotted time. As can be seen, despite slightly worse average rewards, FGFN still outperforms other methods in terms of the number of discovered modes (with the exception of senolytic discovery task, where it fails to discover any high-reward molecules). This includes FGFN+SA, despite its generally worse performance than FGFN. This suggests that RGFN samples are less diverse, possibly due to the relatively small number of fragments and reactions used. However, RGFN still outperforms remaining methods across all tasks, suggesting that it preserves some of the benefits of the diversity-focused GFlowNet framework.

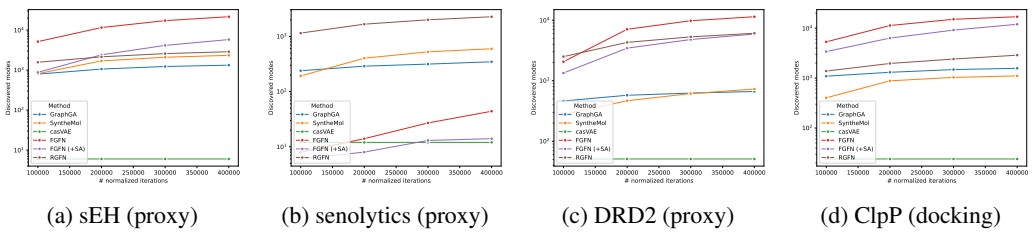

| (a) sEH (proxy) | (b) senolytics (proxy) | (c) DRD2 (proxy) | (d) ClpP (docking) |

Figure 4: Number of discovered modes as a function of normalized iterations. Log scale used.

Finally, we evaluate the synthesizability of the generated compounds as a key output. We present average values of several synthesizability-related metrics, computed over top-k modes generated for each method, in Table 1. We include measures indicating average molecular weight and drug-likeness (QED) to gauge the size of generated compounds. Furthermore, for completeness, we also include SAScores [18], but note that they are only a rough approximation of ease of synthesis. For a better estimate of synthesizability we perform retrosynthesis using AiZynthFinder [23] and count the average number of molecules for which a valid retrosynthesis pathway was found. However, it is important to note that both SAScores and AiZynthFinder scores are inherently noisy metrics. While we evaluate all methods using them for the sake of rigorousness, ultimately molecules generated by RGFN (as well as SyntheMol and casVAE) are guaranteed to be highly likely synthesizable. Note that to reduce variance, we compute molecular weight, QED, and SAScores over the top-500 modes, but due to high computational cost, AiZynthFinder scores are computed only over top-100 modes. As can be seen, while there is some variance across tasks, RGFN performs similarly to SyntheMol and casVAE in terms of both synthesizability scores, and significantly outperforms GraphGA and FGFN. Crucially, including SAScore as a reward does improve the performance of FGFN in terms of that metric, but does not drastically change the AiZynthFinder scores, demonstrating that it is insufficient to guarantee synthesizability. All RGFN modes were additionally inspected manually by an expert chemist and confirmed as synthesizable, which indicates that AiZynth scores are likely underestimated.

Table 1: Average values of synthesizability-related metrics for top-k modes.

| Task | Method | Mol. weight ↓ | QED ↑ | SAScore ↓ | AiZynth ↑ |
|------|--------|---------------|-------|-----------|-----------|
| sEH | GraphGA | 528.6 ± 42.3 | 0.21 ± 0.06 | 3.87 ± 0.24 | 0.04 |
| | SyntheMol | **411.1 ± 66.7** | **0.57 ± 0.18** | 2.85 ± 0.55 | 0.80 |
| | casVAE | 421.6 ± 103.4 | 0.52 ± 0.23 | **2.41 ± 0.47** | **0.82** |
| | FGFN | 473.4 ± 58.9 | 0.39 ± 0.13 | 3.43 ± 0.48 | 0.14 |
| | FGFN+SA | 473.7 ± 62.2 | 0.36 ± 0.12 | 3.01 ± 0.50 | 0.27 |
| | RGFN | 495.2 ± 49.6 | 0.29 ± 0.10 | 3.09 ± 0.39 | 0.56 |
| Seno. | GraphGA | 485.7 ± 75.6 | 0.09 ± 0.05 | 2.92 ± 0.26 | 0.05 |
| | SyntheMol | 441.4 ± 83.5 | 0.48 ± 0.19 | **2.77 ± 0.40** | 0.53 |
| | casVAE | **431.5 ± 100.9** | **0.50 ± 0.19** | 2.82 ± 0.46 | **0.65** |
| | FGFN | 468.9 ± 47.7 | 0.42 ± 0.13 | 3.55 ± 0.52 | 0.02 |
| | FGFN+SA | 451.8 ± 54.5 | 0.32 ± 0.12 | 2.83 ± 0.44 | 0.13 |
| | RGFN | 558.7 ± 62.8 | 0.21 ± 0.09 | 3.24 ± 0.32 | 0.58 |
| ClpP | GraphGA | 521.0 ± 31.8 | 0.32 ± 0.07 | 4.14 ± 0.51 | 0.00 |
| | SyntheMol | 458.2 ± 60.7 | 0.45 ± 0.16 | 2.86 ± 0.56 | 0.56 |
| | casVAE | **423.0 ± 61.7** | **0.47 ± 0.17** | **2.44 ± 0.41** | **0.84** |
| | FGFN | 548.6 ± 42.9 | 0.22 ± 0.03 | 2.94 ± 0.54 | 0.25 |
| | FGFN+SA | 509.2 ± 52.4 | 0.24 ± 0.04 | 2.61 ± 0.49 | 0.33 |
| | RGFN | 526.2 ± 37.6 | 0.23 ± 0.04 | 2.83 ± 0.22 | 0.65 |
| DRD2 | GraphGA | 475.4 ± 53.2 | 0.42 ± 0.12 | 2.50 ± 0.23 | 0.41 |
| | SyntheMol | **365.6 ± 54.3** | **0.72 ± 0.14** | 2.78 ± 0.43 | 0.66 |
| | casVAE | 404.8 ± 83.5 | 0.59 ± 0.20 | 2.42 ± 0.38 | **0.87** |
| | FGFN | 386.5 ± 45.0 | 0.63 ± 0.11 | 2.58 ± 0.54 | 0.76 |
| | FGFN+SA | 381.1 ± 35.1 | 0.64 ± 0.10 | **2.37 ± 0.37** | 0.78 |
| | RGFN | 447.1 ± 45.7 | 0.44 ± 0.10 | 2.79 ± 0.34 | **0.87** |

## 4.3 Scaling to larger sets of fragments

Next we investigate the influence of a fragment embedding scheme proposed in Section 3.2. In the standard implementation of the GFlowNet policy, actions are represented as independent embeddings in the MLP. These encode actions as indices, effectively disregarding their respective structures and all information contained therein. The model must thus select from a library of reagents without any knowledge as to their chemical makeup or properties. While finding similarities between actions may be a relatively easy task for small action spaces, it becomes more difficult when the size of the action space increases. To scale RGFN to a larger size of the building block library, we proposed to encode building block selection actions using molecular fingerprints, allowing the model to leverage their internal structures without any additional computational overhead during inference. In Figure 5, we observe that our fingerprint embedding scheme allows for drastically faster convergence compared to the standard independent action embedding, especially for large library sizes. The details on how the larger fragment libraries were created can be found in Appendix F.

## 4.4 Examination of the produced ligands

In the final stage of experiments we examine the capabilities of RGFN to produce high quality ligands across multiple diverse docking targets (see Appendix G for more details). The aim is to evaluate whether 1) the chemical language used is expressive enough to produce structurally diverse molecules for different targets, and 2) whether the generated ligands form realistic poses in the binding pockets. We first demonstrate the diversity of ligands across targets on a UMAP plot of extended-connectivity fingerprints (Figure 6). Ligands assigned to specific targets form very distinct clusters, showcasing their diversity. Interestingly, we also observe structural differences between sEH proxy and sEH docking, possibly indicating poor approximation of docking scores by the proxy model. Secondly, we examine the docking poses of the highest scoring generated ligands (Figure 7). As can be seen, the generated molecules produce realistic docking poses, closely resembling the poses of known ligands (Appendix K), despite being diverse in terms of structural similarity (Appendix M). We further conduct a cost analysis and synthesis planning for top modes in Appendices N and O. Overall, this demonstrates the usefulness of the proposed RGFN approach in the docking-based screens.

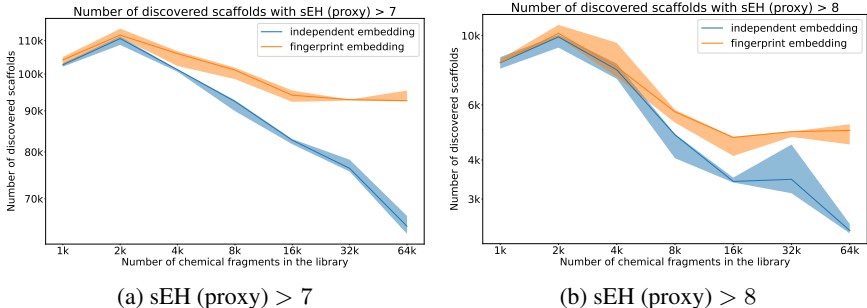

(a) sEH (proxy) > 7            (b) sEH (proxy) > 8

Figure 5: The number of discovered Murcko scaffolds with sEH proxy value above 7 (a) and 8 (b) as a function of fragment library size. We compare standard independent embeddings of fragment selection actions (blue) with our fingerprint-based embeddings (orange) that account for the fragments' chemical structure. The number of scaffolds is reported after 2k training iterations for 3 random seeds (the solid line is the median, while the shaded area spans from minimum to maximum values). We observe that our approach greatly outperforms independent embedding when scaling to a larger action space.

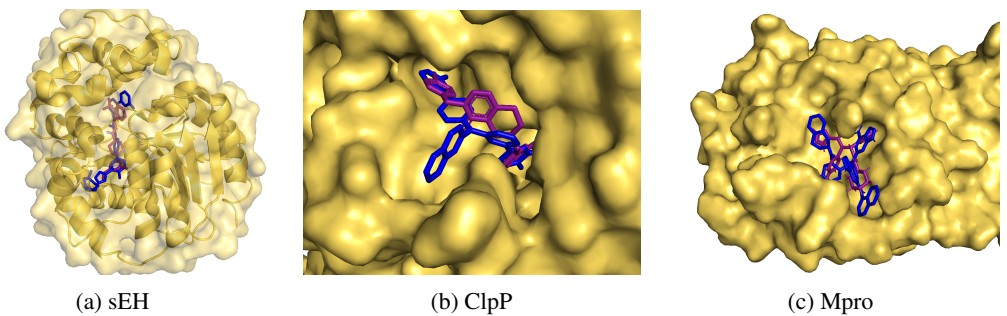

(a) sEH           (b) ClpP           (c) Mpro

Figure 7: Top docked RGFN ligands after filtering steps (blue) overlaid with the PDB-derived ligand (purple) for each of sEH, ClpP, and Mpro.

# 5 Limitations

The current proof-of-principle implementation of RGFN uses only 17 reaction types and 350 building blocks. Although these limited inputs already generate a vast chemical space, this represents only a small fraction of possible drug-like space, which in turn limits the quality and potency of the generated molecular structures. The scaling experiment demonstrates that the number of building blocks can readily be increased, and increasing the number and diversity of building blocks is a straightforward way to enhance and survey the accessible chemical space.

The current set of building blocks and reactions tends to generate linear and flat-shaped molecules. Adding a small set of cyclization reactions (such as peptide macrocyclization and ring-closing metathesis), along with more complex-shaped scaffold building blocks, as well as reactions that introduce sp$^3$ hybridized atoms and stereochemical complexity will therefore allow for greater shape diversity and the

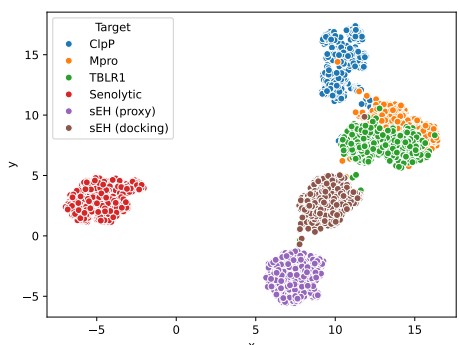

Figure 6: UMAP plot of chemical structures of top-500 modes generated for each target. RGFN generates sufficient chemical diversity to produce distinct clusters of compounds. See Appendix G for description of each target protein.

generation of more potent molecules [51, 20]. It is also important to recognize that RGFN does not explicitly generate synthetic routes to the molecules, at least not in a strict chemical sense, which in addition to a sequence of reactions transforming sets of reactants into products (which RGFN does provide), would also include choice of reaction conditions, external reagents, catalysts, protection group strategies, etc.

Another significant limitation in the quality of the generated molecular structures is the reliance on molecular docking as a scoring oracle. Although molecular docking has been successfully used in large-scale virtual screening efforts [44, 61, 35], it has well-known shortcomings in its predictive power. First, docking scores correlate strongly with molecular weight (MW) [13] and do not account for drug-likeness requirements like optimal MW or ClogP. In this work, molecule size was constrained only by the number of reaction steps, encouraging RGFN to generate large molecules within the building block limit. This can be somewhat rectified by augmenting reward with a drug-likeness or ligand efficiency term. Second, its binding affinity predictions and rankings often correlate weakly with experimental values and are highly dependent on the nature of the target protein's binding site [73, 71]. This is further illustrated by the fact that known ligands are not necessarily characterized by highest possible docking scores (Appendix L). This limitation impacts the learning of the chemical structure-activity relationship space, leading to the generation of sub-optimal molecules. One solution to this limitation is to incorporate more accurate but computationally expensive methods (such as ensemble docking, MM-PBSA, and FEP) within a multi-fidelity framework [26]. However, since we focus on robust, affordable, and facile synthesis methods, we ultimately aim to extend our approach beyond computational scoring methods by directly conducting experimental evaluation of synthesized compound batches within an active learning loop.

## 6    Conclusions

In this paper, we present RGFN, an extension of the GFlowNet framework that operates in the action space of chemical reactions. We propose a curated set of high-yield chemical reactions and low-cost molecular building blocks that can be used with the method. We demonstrate that even with a small set of reactions and building blocks, the proposed approach produces a state space with a size orders of magnitude larger than typical experimental screening libraries while ensuring high synthesizability of the generated compounds. We also show that the size of the search space can be further increased by including additional building blocks and that the proposed action embedding mechanism improves scalability to very large building block spaces.

In the course of our experiments, we show that RGFN achieves roughly comparable average rewards to state-of-the-art methods, and it outperforms another approach operating directly in the space of chemical reactions and, crucially, standard fragment-based GFlowNets. At the same time, it significantly improves the synthesizability of generated compounds when compared to a fragment-based GFlowNet. Analysis of ligands produced across the set of diverse tasks demonstrates sufficient diversity of proposed chemical space to generalize to various targets. While not yet demonstrated experimentally, ease of synthesis (due to the small stock of cheap fragments and high-yield chemical reactions used) combined with reasonably high optimization quality of bespoke ligands offer a promising alternative to standard high-throughput screening applications. In particular, it can be beneficial for active learning-based pipelines with significant wet lab component, reducing the reliance on inaccurate docking oracles. Facilitating the drug discovery process through the generation of novel small molecules can eventually lead to the discovery of novel medications leading to significant societal benefits.

## Acknowledgments and Disclosure of Funding

This work was supported by funding from CQDM Fonds d'Accélération des Collaborations en Santé (FACS) / Acuité Québec and the National Research Council (NRC) Canada, the Canadian Institutes for Health Research (CIHR), grant no. FDN-167277, Samsung and Microsoft. D. Shevchuk is grateful for support from the Mitacs Globalink Graduate Fellowship, grant no. FR121160/FR121161. The research of P. Gaiński was supported by the National Science Centre (Poland), grant no. 2022/45/B/ST6/01117. Additional thanks for funding provided to the University of Toronto's Acceleration Consortium from the Canada First Research Excellence Fund, grant no. CFREF-2022-00042. Computational resources were provided by the Digital Research Alliance of

Canada (`https://alliancecan.ca`) and Mila (`https://mila.quebec`). We gratefully acknowledge Poland's high-performance Infrastructure PLGrid (ACK Cyfronet Athena, HPC) for providing computer facilities and support within computational grant no PLG/2023/016550.

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

# A    State space size estimation

We estimate the state space size by first sampling 1,000 random trajectories, masking out the end-of-sequence action unless the maximum trajectory length $max$ is reached. Then, for every $i$-th reaction or fragment in the trajectory, we count the average number of valid fragments $frag_i$ and reactions $react_i$ from a given state in the trajectory, as well as the average number of unique trajectories $traj_i$ into which a state can be decomposed using the backward policy. We estimate the state space size as

$$\frac{(\prod_{i=0}^{max} frag_i)(\prod_{i=1}^{max} react_i)}{traj_{max}}. \tag{9}$$

Experimentally derived average values of these parameters can be found in Table 2. Note that in the second setting we randomly picked 8,000 fragments from the Enamine stock (with the same balancing procedure as in Section 4.3), which after merging with our own fragments, canonization and duplicate removal yielded a total of 8,317 fragments.

Table 2: Experimentally derived average values of valid fragments, valid reactions, and possible trajectories.

|  | 350 fragments | | | 8350 fragments | | |
|---|---|---|---|---|---|---|
| # reactions | $frag_i$ | $react_i$ | $traj_i$ | $frag_i$ | $react_i$ | $traj_i$ |
| 0 | 350.0 | - | 1.0 | 8317.0 | - | 1.0 |
| 1 | 37.5 | 11.8 | 3.5 | 835.8 | 12.0 | 4.2 |
| 2 | 39.9 | 16.5 | 16.8 | 822.1 | 15.9 | 17.0 |
| 3 | 40.7 | 15.4 | 76.7 | 832.6 | 17.0 | 75.2 |
| 4 | 40.0 | 15.8 | 349.3 | 814.0 | 18.0 | 480.8 |
| 5 | 42.1 | 16.8 | 1825.8 | 857.6 | 18.9 | 3058.1 |

# B    Training details

## B.1    Proxy models

The sEH proxy is described in [4]. It is an MPNN trained on a normalized docking score data. We utilize the exact same model checkpoint as provided in [57].

The senolytic classification model is a graph neural network trained on the biological activity classification task of senolytic recognition [74]. Specifically, it was trained on two combined, publicly available senolytic datasets [74, 65]. Reward is given by the predicted probability of a compound being a senolytic. It is worth noting that due to the low amount of data and high imbalance ($< 100$ active compounds, a high proportion of which contained macrocycles and were infeasible to construct with fragment-based generative models), this is expected to be a difficult task with sparse reward.

The senolytic proxy model is GNEprop [62], which consisted of 5 GIN layers [75] with hidden dimensionality of 500, utilized Jumping Knowledge shortcuts [76], and had a single output MLP layer. Pretraining was done in an unsupervised fashion on the ZINC15 dataset [66]. The training was done for 30 epochs using the Adam optimizer with a learning rate of $5 \times 10^{-5}$ and batch size of 50.

The DRD2 proxy is described in [53, 28]. It is a support vector machine classifier with a Gaussian kernel using ECFP6 fingerprints as a feature representation.

## B.2    Generative models

Both RGFN and FGFN were trained with trajectory balance loss [45] using Adam optimizer with a learning rate of $1 \times 10^{-3}$, logZ learning rate of $1 \times 10^{-1}$, and batch size of 100. The training lasted 4,000 steps. A random action probability of 0.05 was used, and RGFN used a replay buffer of 20 samples per batch. Both methods use a graph transformer policy with 5 layers, 4 heads, and 64 hidden dimensions. Exponentiated reward $R(x) = exp(\beta * score(x))$ was used, with $\beta$ dependent on the task: 8 for sEH proxy, 0.5 for senolytic proxy, 48 for DRD2 proxy, and 4 for all docking runs. Note

that due to different ranges of score values, this resulted in a roughly comparable range of reward values.

For FGFN+SA, we used a modified reward function $R(x) = exp(\beta * (0.5 * proxy(x)/max\_proxy + (10 - SA\_score)/10)$, with $\beta$ adjusted per proxy to match the original reward range.

All sampling algorithms were outfitted with the Vina GPU-2.1 docking, senolytic proxy, sEH proxy, and DRD2 proxy scoring functions. While model architecture hyperparameters and batch sizes were kept consistent between FGFN and RGFN, we allowed FGFN a maximum fragment count of 6 as opposed to RGFN's 5 due to RGFN's larger average building block sizes.

GuacaMol's Graph GA model was trained with a population size of 100, offspring size of 200, and a mutation rate of 0.01 for 2000 generations for a total of 400,000 visited molecules.

For SyntheMol experiments, we used the default building block library of 132,479 compatible molecules and pre-computed docking, senolytic, and sEH proxy scores for all prior to executing rollouts to follow the established methodology. Due to CPU constraints, sampling 500,000 molecules with SyntheMol was impractical. Instead, we executed 100,000 rollouts over approximately 72 hours to match the RGFN training time with docking, yielding 111,964 unique molecules. Additionally, we performed 50,000 rollouts each (approximately 24 hours) for sEH, senolytic, and DRD2 proxies, resulting in 73,941, 69,652, and 62,320 unique molecules, respectively.

casVAE was trained with Bayesian optimization (BO) using default parameters consisting of a hidden size of 200, latent size of 50, and message passing depth of 2. Again, due to time constraints imposed by BO latent space updates, we instead opted to approximately match RGFN training times, training for 72, 24, 24, and 24 hours on the docking, sEH, senolytic, and DRD2 tasks respectively for a total of 135, 41, 38, and 40 rollouts and 7708, 2097, 1521, and 2165 total generated molecules, respectively.

## C  GPU-accelerated docking

Our docking oracle first accepts canonized SMILES strings as input. These are then converted to RDKit Molecules, protonated, and a low-energy conformer is generated and minimized with the ETKDG [58] conformer generation method and UFF force field [56], respectively. For computational efficiency, we generate one initial conformer per ligand. Each conformer is converted to a pdbqt file and docked against a target with Vina-GPU 2.1 using model defaults: exhaustiveness (denoted by "thread" in the implementation) of 8000 and a heuristically determined search depth $d$ given by

$$d = \max\left(1, \lfloor 0.36 \times N_{atom} + 0.44 \times N_{rot} - 5.11 \rfloor\right), \tag{10}$$

where $N_{atom}$ and $N_{rot}$ are the number of atoms and the number of rotatable bonds, respectively, in the generated molecule. Box sizes were determined individually to encompass each target binding site and centroids were calculated to be the average position of ligand atoms in the receptor template PDB structure file. A negative score is calculated and returned as a reward.

### C.1  Target preprocessing

Each target was prepared by removing its complexed inhibitor and atoms of other solvent or solute molecules. We selectively prepared the ClpP 7UVU protein structure by retaining only two monomeric units to ensure the presence of a single active site available for ligand binding and similarly prepared the Mpro 6W63 protein structure by retaining only one monomeric unit.

### C.2  Boron substitution for docking

Due to the lack of force field parameters for boron atoms, QuickVina2 is unable to process ligands with boronic acid or ester groups. Therefore, we chose to substitute the boronic acid group and its derivatives in the building blocks/reactants with a carboxylic acid, where the carbonyl carbon is a C13 isotope. This was done for two reasons: firstly, carboxylic acids are considered to be bioisosteric to boronic acids[69], and secondly, the $C^{13}$ tag would allow us to distinguish actual carboxylic acids from boronic acid surrogates for SMARTS encoding purposes. This allowed us to dock all possible final products while maintaining structural similarity to the original group and specificity to compatible reaction SMARTS.

## D    Computational resources used

Evaluating fragment scaling took approximately 800 GPU hours on GeForce RTX 4090 in total. Remaining experiments took roughly 24 GPU hours per run on Quadro RTX 8000 for sEH, DRD2 and senolytic proxies, and roughly 72 GPU hours per run on an A100 for docking-based proxies.

## E    Implicit reactions

In this work, we define implicit reactions as SMARTS-encoded reactions, products of which contain atoms that are not included in the reactants and come from "implicit reagents". One example is urea synthesis using carbonyl surrogates: the SMARTS template uses two amines, but the product has more atoms than specified in the reactants. These come from the "implicit reagent", in our case — any phosgene surrogate (e.g., CDI) (Figure 8).

Figure 8: An example of an implicit reaction: urea synthesis reaction encoded in SMARTS.

In addition to urea formation, other implicit reactions encoded in SMARTS include azide-alkyne cycloaddition, azide-nitrile cycloadditions, and tetrazole synthesis using peptide terminal thiourea cyclizations.

## F    Sampling large fragment set from Enamine building blocks

In order to evaluate our approach at scale, our building block set had to be balanced to prevent introducing learning biases via building block selection. This was achieved by grouping reagents into twelve fragment classes, for which a specific weight was assigned based on the frequency of appearance of these structures in different reactions. For example, a carboxylic acid appearing in two distinct reactions was assigned a weight coefficient of two, which would later be used to calculate the normalized amount of building blocks per group for a specific database size using the following formula:

$$\text{Amount of BBs} = \frac{\text{Weight coef.}}{\text{Sum of weights}} * \text{database size} \tag{11}$$

The only exception is Michael acceptors, or group 3, which has a weight of 0.25 due to the low availability of these reagents in most public molecular databases and possible overlap with structures that might fit the corresponding SMARTS code (for example, 2-cyano or 2-carbonyl aromatic compounds).

The final set of fragments was constructed by randomly sampling fragments from publicly available Enamine building blocks in a way that roughly preserved the above grouping. This was done in a greedy round-robin fashion, randomly selecting one fragment at a time from the remaining fragments belonging to a given group, and iterating through all groups until the specified number of fragments was selected.

## G    Target selection

Our targets (sEH, Mpro, ClpP, and TBLR1) were chosen with diverse functions and binding sites in mind as test cases for RGFN. ClpP is a highly conserved compartmentalized protease that degrades

substrates in a signal-dependent fashion and is a promising target in cancer and infectious disease [6]. A number of structurally unrelated small molecules that allosterically activate ClpP, thereby deregulating its protease activity, inhibit the growth of cancer cells and bacterial pathogens [31, 25, 30].

The main protease of SARS-CoV-2, Mpro (also called 3CLpro), is required for processing of the virally-encoded polyprotein and thus for viral replication, and is a well-validated and intensively studied anti-SARS-CoV-2 target [14]. The Mpro catalytic pocket contains 4 distinct sub-pockets that recognize amino acid motifs in substrate sites and is a challenging target due to its structural plasticity [36].

Soluble epoxide hydrolase (sEH) catalyzes a key step in the biosynthesis of eicosanoid inflammation mediators and is being pursued as potential target in cardiovascular and other diseases [16]. sEH has been used as a benchmark substrate for machine learning-based drug discovery and is particularly amenable to computational methods because of its deep hydrophobic pocket [47].

Finally, TBL1 and its paralog TBLR1 are components of the NCoR transcriptional repressor complex and contain WD40 domains that participate in protein interactions [78]. TBL1/TBLR1 mediates the transcriptional repression function of the MeCP2 methylCpG-binding protein, which is mutated in the neurodevelopmental disorder Rett syndrome. Loss of function mutations in Rett syndrome frequently disrupt the interaction of a disordered region in MeCP2 that binds to the WD40 domain of TBL1/TBLR1; conversely, MeCP2 gain-of-function by copy number mediated overexpression leads to X-linked intellectual disability [63].

Well-validated experimental assays are available for each of these 4 different targets [31, 14, 47, 1].

# H   Ligand post-processing

To ensure the diversity, specificity, and conformer validity of top-generated molecules for each target, we initially categorized our molecules into distinct modes, each representing any SMILES string with a Tanimoto similarity of 0.5 or lower with all other modes. Subsequently, we selected the top 100 modes based on their Vina-GPU 2.1 scores and filtered their docked poses using PoseBusters [11] in "mol" mode, where any pose failing any PoseBusters check was excluded from consideration. As a final precaution, we selected only modes with Tanimoto coefficients to known aggregators of 0.4 or lower using UCSF's Aggregation Advisor [29] dataset. This process resulted in 35, 68, 31, and 15 top modes for sEH, ClpP, Mpro, and TBLR1 binders, respectively Appendix J. Comparative analyses of docked top RGFN modes and confirmed sEH, ClpP, and Mpro ligand poses can be found in Appendices K to M.

# I   Posebusters, Aggregation Advisor analysis of top 100 modes

Table 3: Proportion of top-100 generated sEH, ClpP, Mpro, and TBLR1 modes satisfying each of PoseBuster's Mol-mode checks, as well as the Aggregation Advisor Tanimoto similarity threshold of 0.4. Molecules were assessed in PoseBuster according to their ability to be loaded and sanitized in RDKit, reasonableness of bond lengths and angles, lack of steric clashes in the pose, aromatic ring and double bond flatness, and internal energy.

| Target | Condition | | | | | | | | | |
| | Load | Sanitize | Connected | Bond Lengths | Bond Angles | Steric Clashes | Aromatic Ring Flatness | Double Bond Flatness | Internal Energy | Agg. Sim. < 0.4 |
| --- | --- | --- | --- | --- | --- | --- | --- | --- | --- | --- |
| sEH | 1.0 | 1.0 | 1.0 | 1.0 | 1.0 | 1.0 | 1.0 | 1.0 | 0.9 | 0.38 |
| ClpP | 1.0 | 1.0 | 1.0 | 1.0 | 1.0 | 0.98 | 1.0 | 1.0 | 0.94 | 0.72 |
| Mpro | 1.0 | 1.0 | 1.0 | 1.0 | 1.0 | 0.89 | 1.0 | 1.0 | 0.62 | 0.65 |
| TBLR1 | 1.0 | 1.0 | 1.0 | 1.0 | 1.0 | 0.96 | 1.0 | 1.0 | 0.34 | 0.53 |

# J   Top filtered molecules for all targets

**Senolytics**

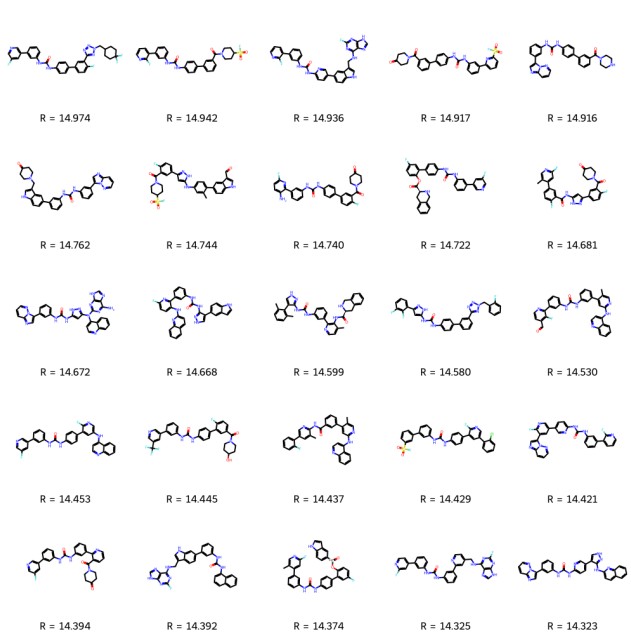

Figure 9: Top 25 senolytic compound modes generated by RGFN.

**sEH**

Figure 10: Top 25 filtered binders to sEH drawn from top 100 RGFN modes.

**ClpP**

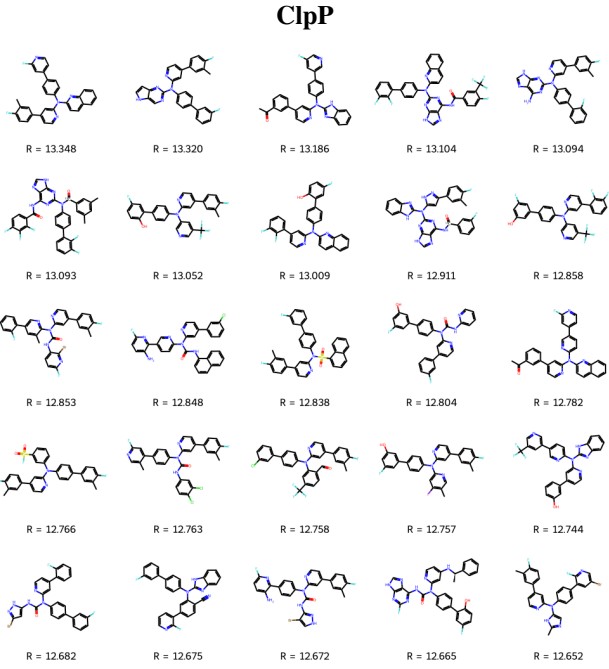

Figure 11: Top 25 filtered binders to ClpP drawn from top 100 RGFN modes.

**Mpro**

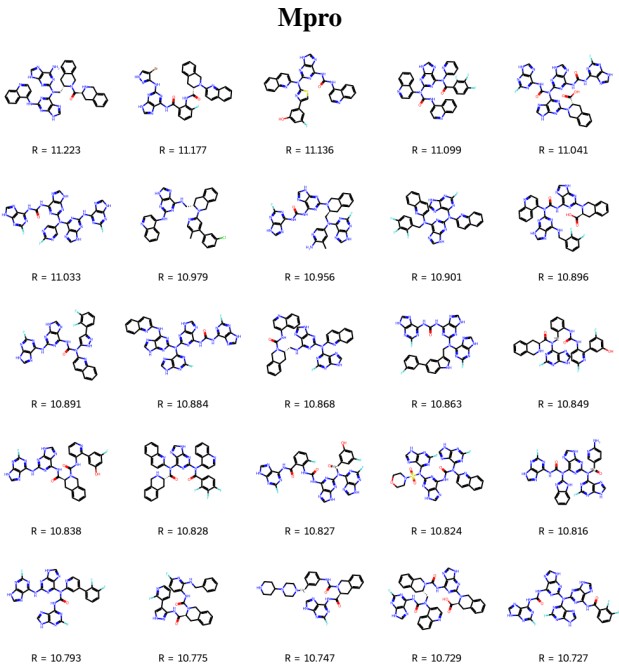

Figure 12: Top 25 filtered binders to Mpro drawn from top 100 RGFN modes.

**TBLR1**

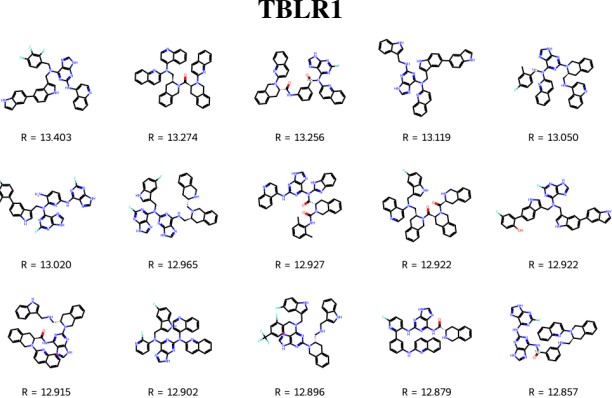

Figure 13: All 15 filtered binders to TBLR1 drawn from top 100 RGFN modes.

# K Docked poses of top generated molecules

**Ours (sEH, 4JNC)**

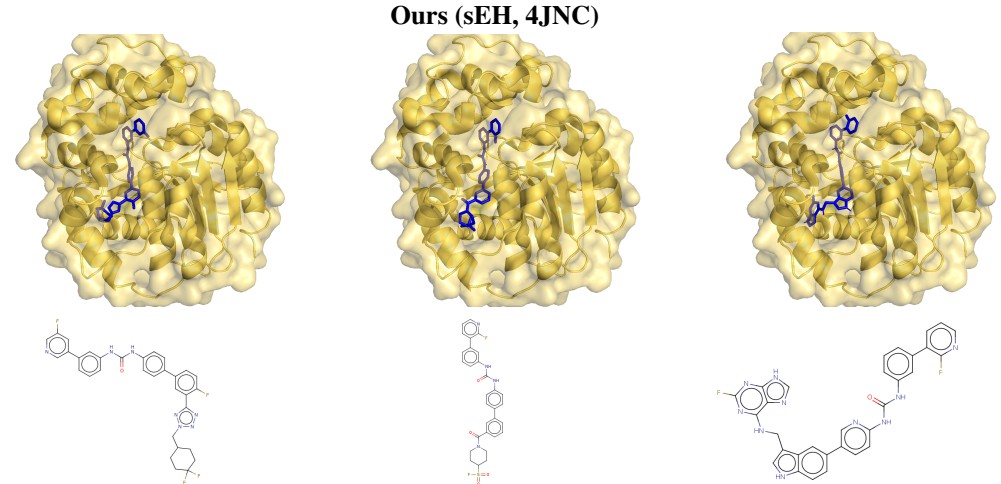

Vina-GPU 2.1 Score: -14.97 Vina-GPU 2.1 Score: -14.94 Vina-GPU 2.1 Score: -14.94

**Reference (sEH, 4JNC)**

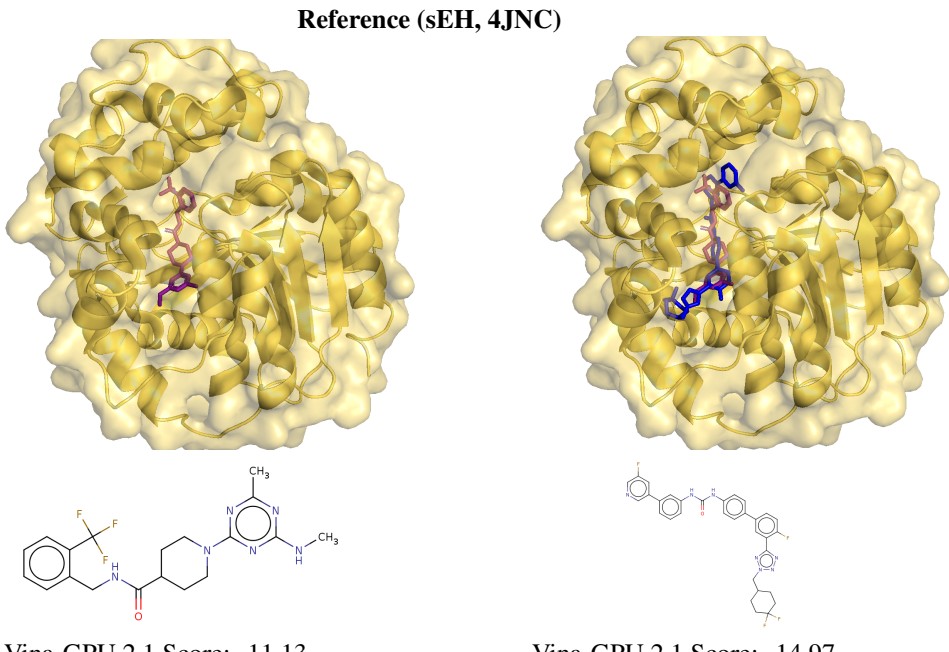

Vina-GPU 2.1 Score: -11.13 Vina-GPU 2.1 Score: -14.97

Figure 14: Top left to right: Top 3 generated ligand scaffolds for sEH (blue). Bottom left: Reference ligand pose (purple, PDB ID: 1LF). Bottom right: Reference ligand (purple) overlaid with top-scoring ligand (blue).

**Ours (ClpP, 7UVU)**

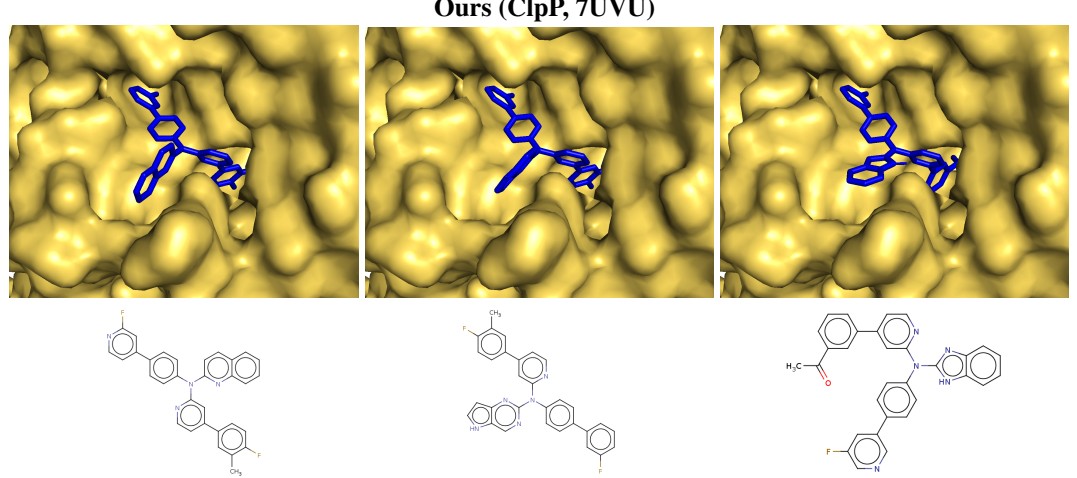

Vina-GPU 2.1 Score: -13.35          Vina-GPU 2.1 Score: -13.32          Vina-GPU 2.1 Score: -13.19

**Reference (ClpP, 7UVU)**

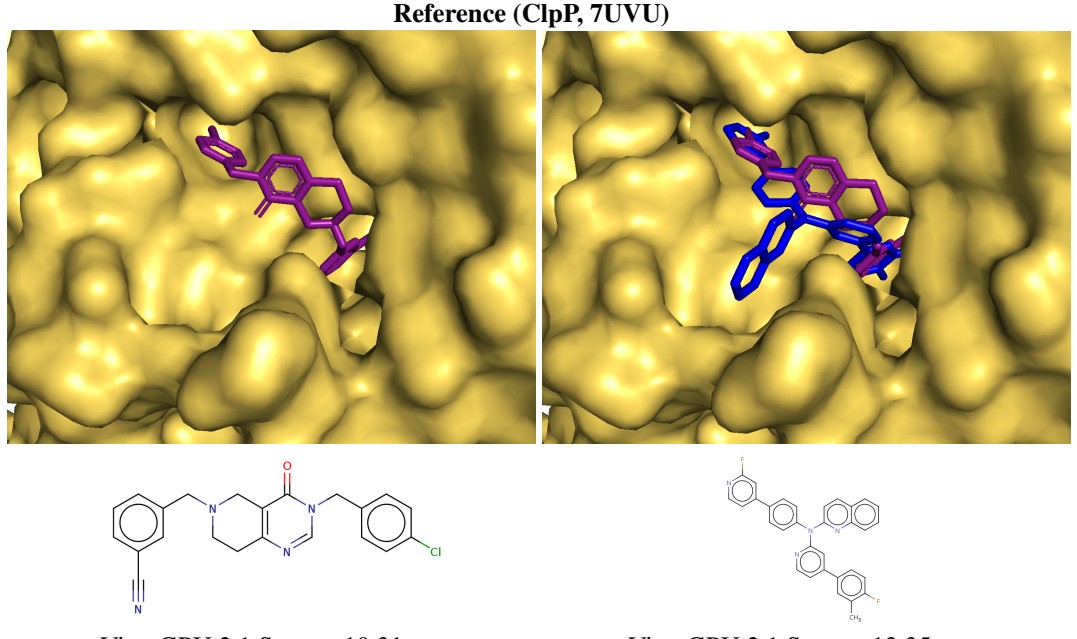

Vina-GPU 2.1 Score: -10.31                    Vina-GPU 2.1 Score: -13.35

Figure 15: Top left to right: Top 3 generated ligand scaffolds for ClpP (blue). Bottom left: Reference ligand pose (purple, PDB ID: OY9). Bottom right: Reference ligand (purple) overlaid with top-scoring ligand (blue).

**Ours (Mpro, 6W63)**

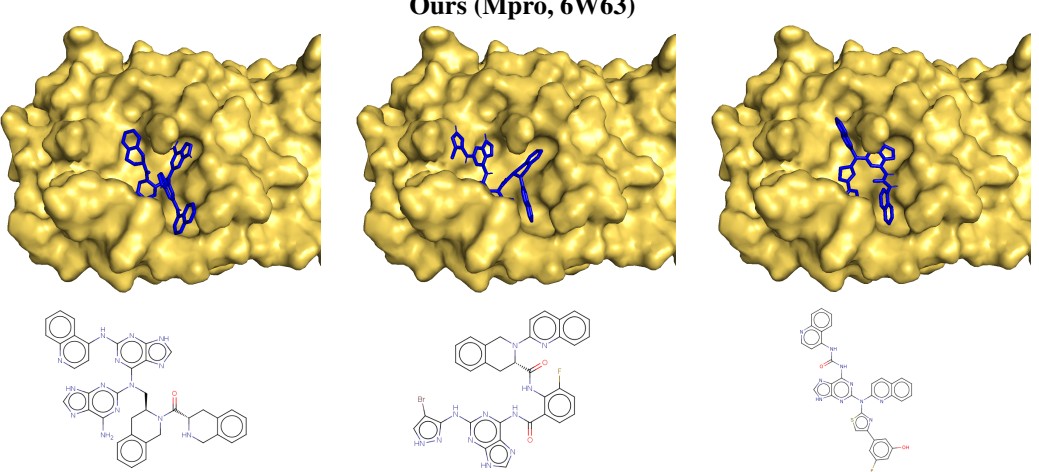

Vina-GPU 2.1 Score: -11.22     Vina-GPU 2.1 Score: -11.18     Vina-GPU 2.1 Score: -11.14

**Reference (Mpro, 6W63)**

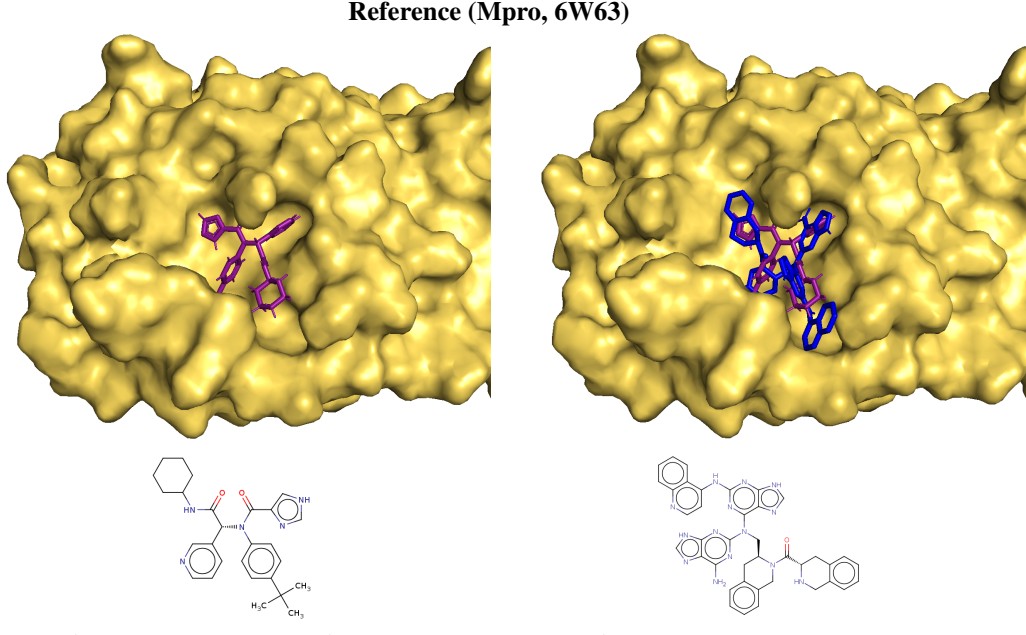

Vina-GPU 2.1 Score: -8.53                    Vina-GPU 2.1 Score: -11.22

Figure 16: Top left to right: Top 3 generated ligand scaffolds for Mpro (blue). Bottom left: Reference ligand pose (purple, PDB ID: X77). Bottom right: Reference ligand (purple) overlaid with top-scoring ligand (blue).

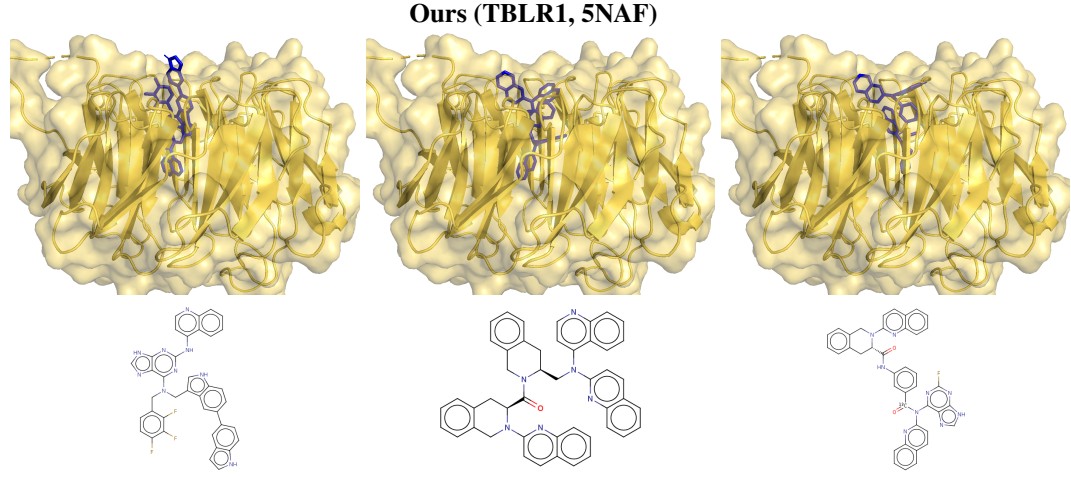

**Ours (TBLR1, 5NAF)**

Vina-GPU 2.1 Score: -13.40     Vina-GPU 2.1 Score: -13.27     Vina-GPU 2.1 Score: -13.26

Figure 17: Left to right: Top 3 generated ligand scaffolds for the TBLR1 WD40 domain. TBLR1 has no known small molecule ligands.

## L    Docking analysis of known ligands

Table 4: Vina-GPU 2.1 docking scores of 10 PDB-available ligands per target.

| Ligand | sEH PDB ID | Score | ClpP PDB ID | Score | Mpro PDB ID | Score |
|---|---|---|---|---|---|---|
| 1 | 5IV | -11.79 | OX0 | -11.10 | J7O | -10.13 |
| 2 | XDZ | -11.64 | PJF | -10.67 | KAE | -9.45 |
| 3 | WJ5 | -11.30 | P4I | -10.40 | 7YY | -9.33 |
| 4 | E3N | -11.14 | P3O | -10.39 | 7XB | -8.6 |
| 5 | 1LF | -11.13 | OY9 | -10.31 | XYV | -8.6 |
| 6 | 8S9 | -11.07 | ZLL | -10.18 | X77 | -8.54 |
| 7 | TK9 | -9.64 | 7SR | -10.17 | XF1 | -8.40 |
| 8 | G3W | -9.60 | OSR | -9.49 | 0EN | -8.19 |
| 9 | G3Q | -9.14 | ONC | -9.47 | J7R | -8.00 |
| 10 | J0U | -8.75 | 9DF | -9.39 | 4N0 | -7.33 |

We calculate and report the Vina-GPU 2.1 docking scores of 10 true ligands per target assessed in the paper. Each PDB was prepared as described in Appendix C.1.

# M Tanimoto similarity to known ligands

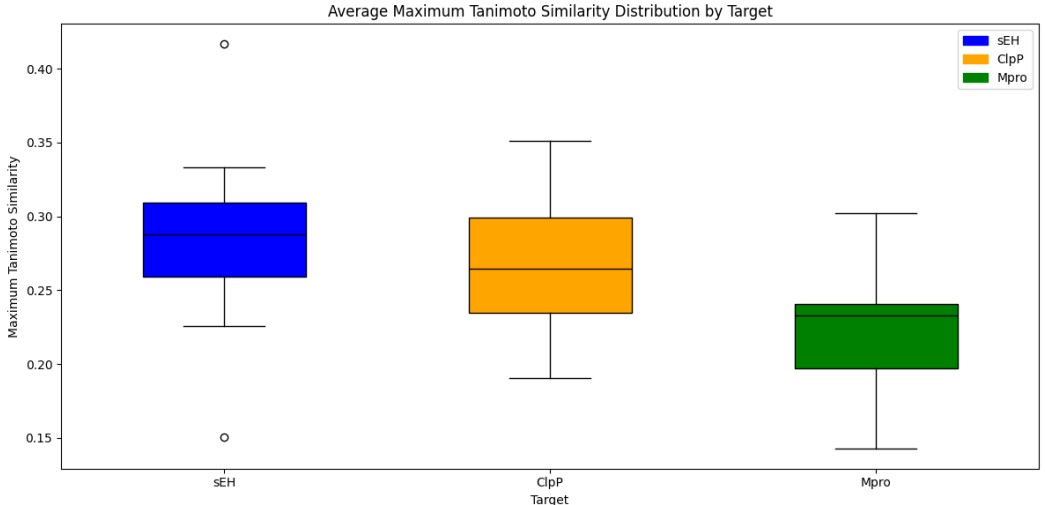

Figure 18: Average Maximum Tanimoto Similarity to known ligands. For each target, we plot the highest Tanimoto similarity score across all generated molecules to the ten corresponding PDB-derived ligands.

# N    Ligands cost analysis

To demonstrate the synthesizability and cost-effectiveness of the ligands produced by RGFN, we conducted a comprehensive cost analysis and proposed a synthesis plan for the top 10 scored ClpP hits generated by RGFN and SyntheMol (see Appendix O). We compared these two methods due to their similarity in chemical approach. For the cost calculation, we only considered the price of building blocks, which were directly sourced from EnamineStore for the case of Synthemol ligands. All prices are represented in US dollars per 0.1 mmol of product (Figures 19 and 20). Notably, for very common reagents only available in bulk, costs per gram were estimated by taking the cheapest or smallest available alternative and dividing its cost by its mass. All prices for individual building blocks were sourced from their respective vendors, which included Millipore Sigma, Oakwood Chemicals, Combi-Blocks, TCI, and AngeneSci.

| Position | Ligand | Cost per 0,1 mmol, $ | Position | Ligand | Cost per 0,1 mmol, $ |
|---|---|---|---|---|---|
| 1 | | 2.07 | 6 | | 1.72 |
| 2 | | 1.90 | 7 | | 3,93 |
| 3 | | 1.76 | 8 | | 1.82 |
| 4 | | 1.37 | 9 | | 2.35 |
| 5 | | 1.80 | 10 | | 1.84 |

Figure 19: Synthesis cost for top-10 scoring ClpP ligands produced by RGFN.

| Position | Ligand | Cost per 0,1 mmol, $ | Position | Ligand | Cost per 0,1 mmol, $ |
|---|---|---|---|---|---|
| 1 | | 185.79 | 6 | | 112.84 |
| 2 | | 17.68 | 7 | | 263.95 |
| 3 | | 260.33 | 8 | | 78.62 |
| 4 | | 233.44 | 9 | | 35.44 |
| 5 | | N/A | 10 | | 185.02 |

Figure 20: Synthesis cost for top-10 scoring ClpP ligands produced by Synthemol.

As expected, ligands generated by RGFN are significantly cheaper to produce compared to SyntheMol, despite having more synthesis steps and lower overall theoretical yields, with an average of 55-70% after 4 steps in the case of RGFN compared to 90-95% average yield after 1 step for SyntheMol. Noticeably, SyntheMol ligand 5 wasn't found to be synthesizable by using reactions proposed in their work. After conducting further analysis, it was found that such a compound could have been produced by Reaction 8 via nucleophilic substitution of the fluorine atom in the trifluoromethyl group; however, this particular reaction is not likely to be performed with sufficient yields, and therefore such a product cannot be considered synthesizable.

## O    Examples of plausible synthetic routes to molecules

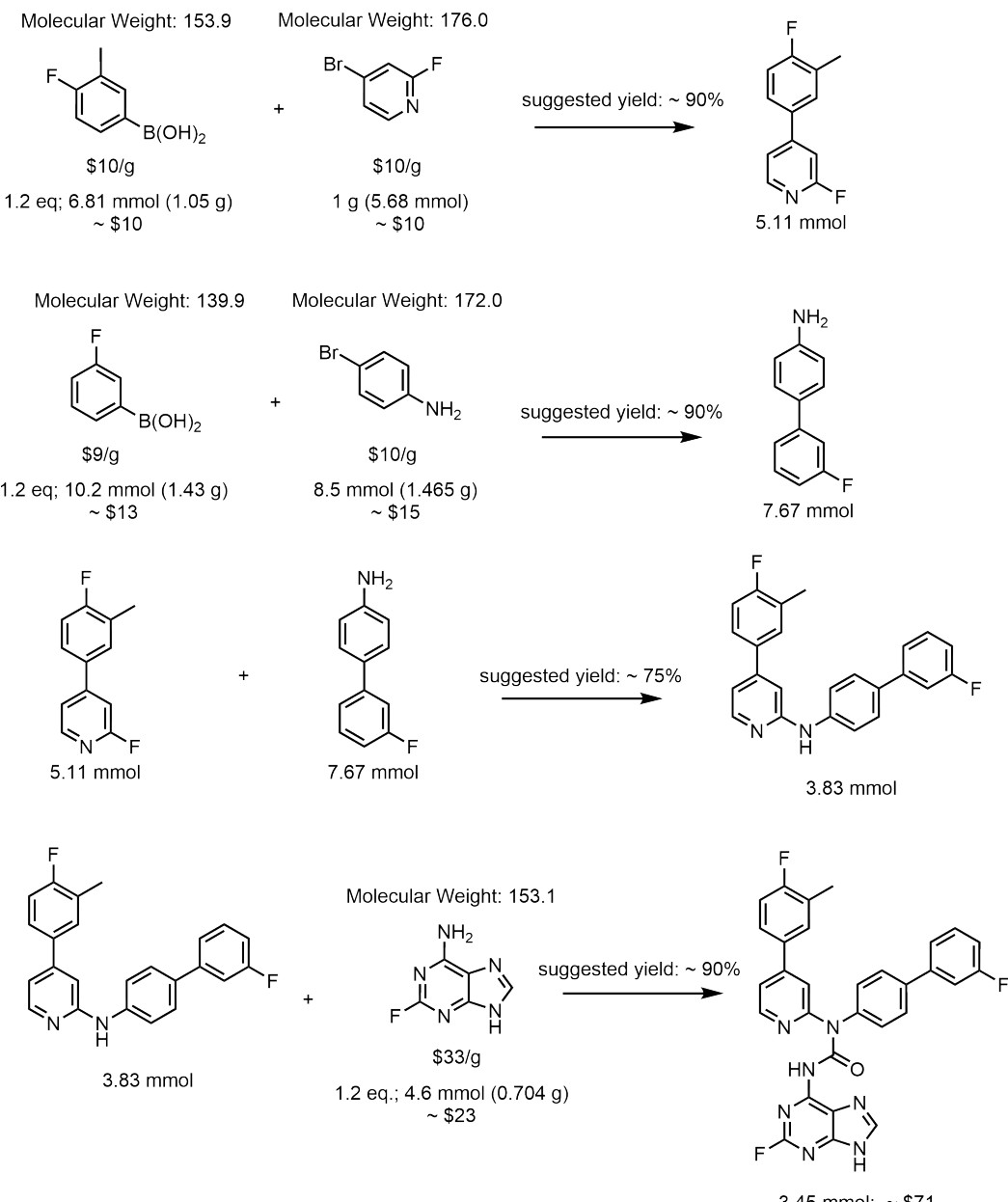

Figure 21: Plausible synthesis plan and estimated precursor cost for RGFN-produced ClpP ligand 1.

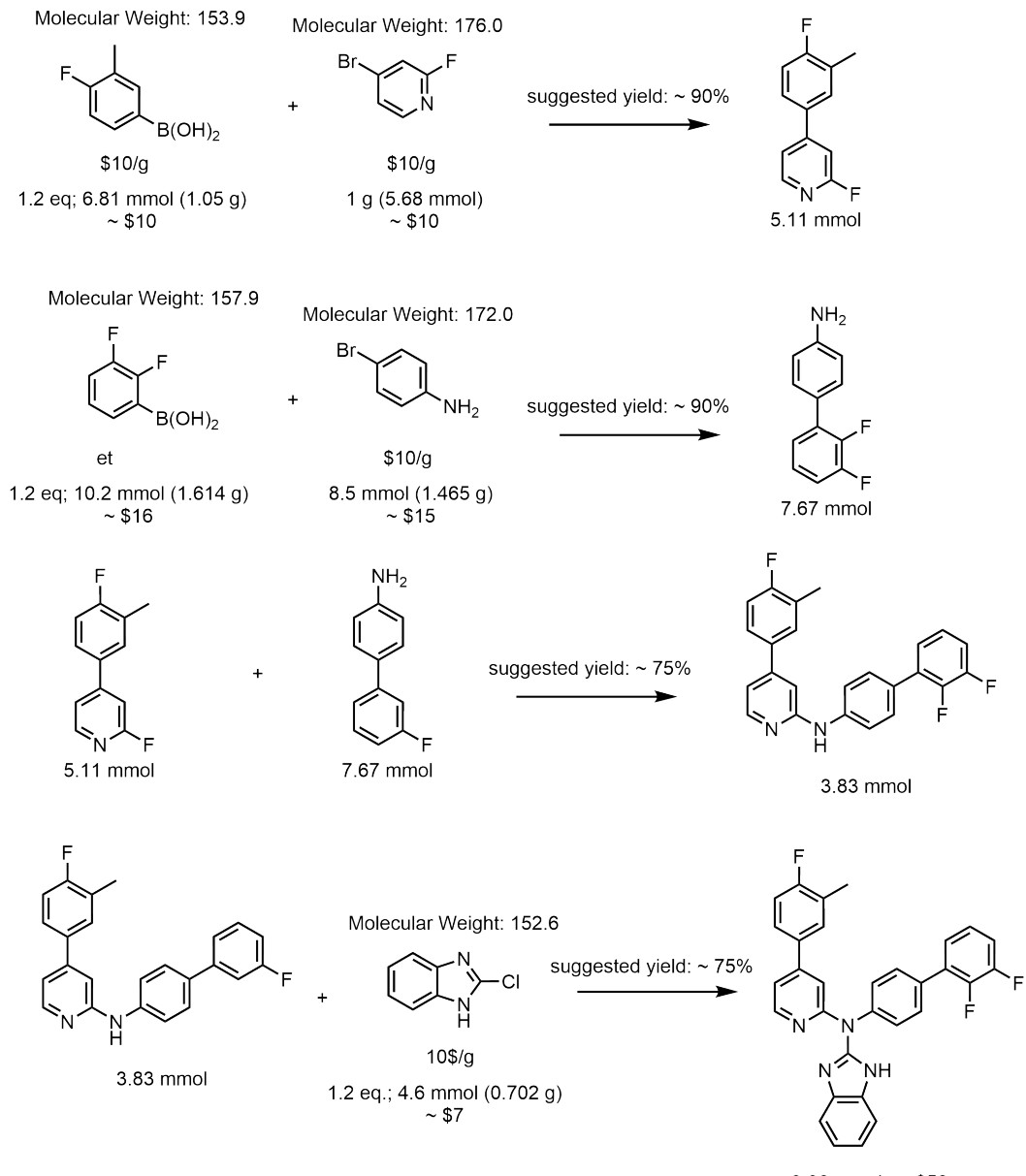

Figure 22: Plausible synthesis plan and estimated precursor cost for RGFN-produced ClpP ligand 2.

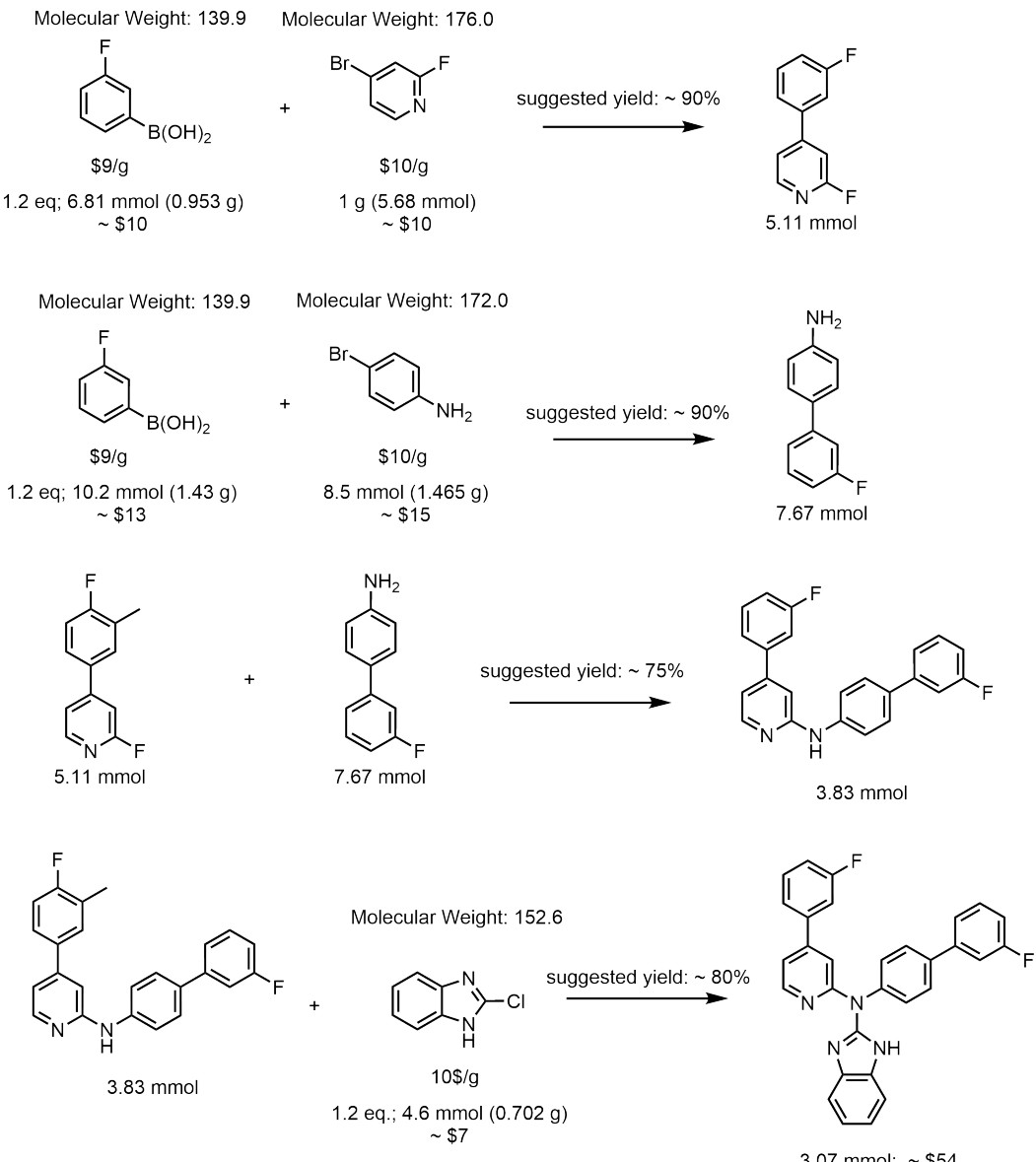

Figure 23: Plausible synthesis plan and estimated precursor cost for RGFN-produced ClpP ligand 3.

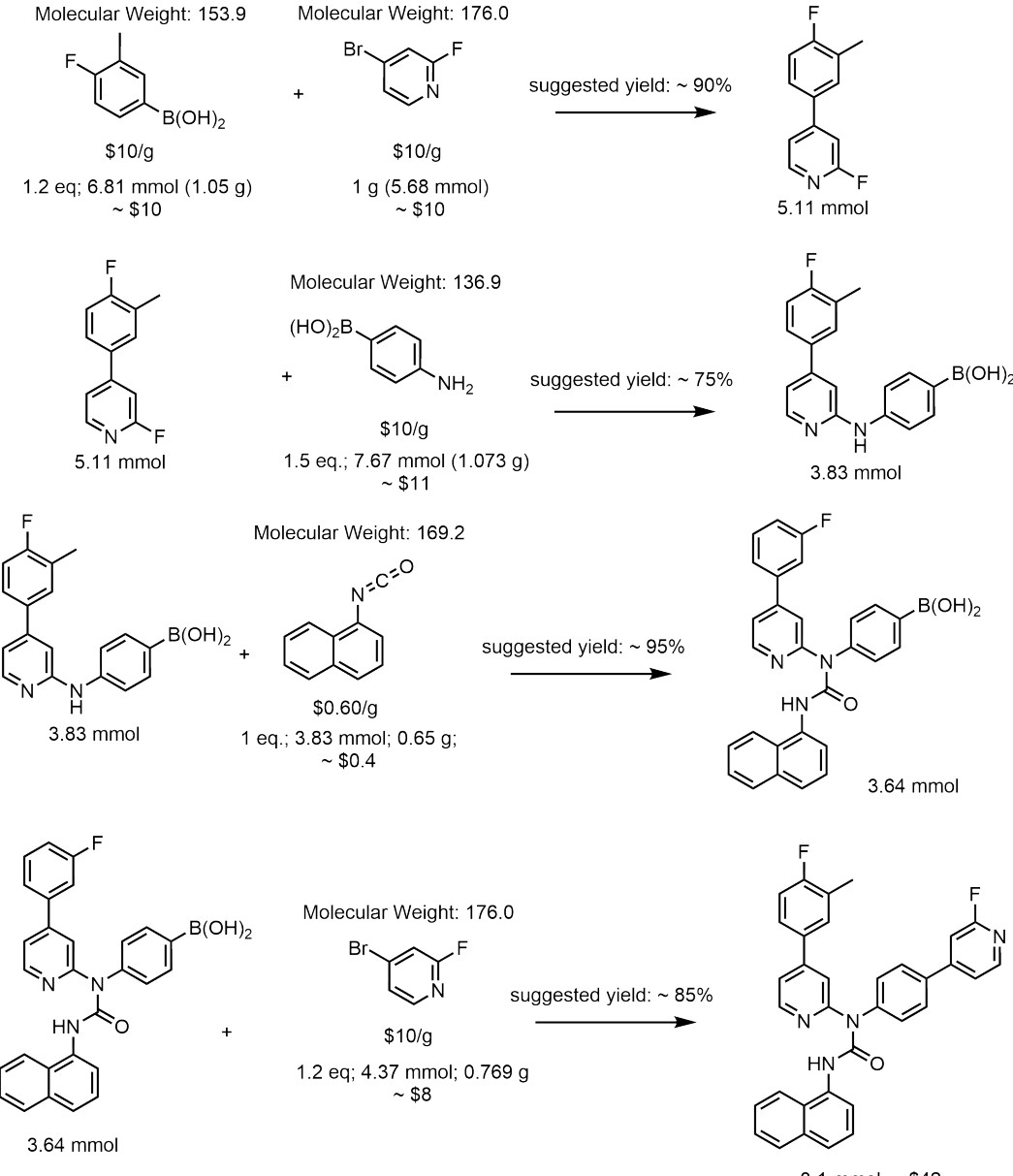

Figure 24: Plausible synthesis plan and estimated precursor cost for RGFN-produced ClpP ligand 4.

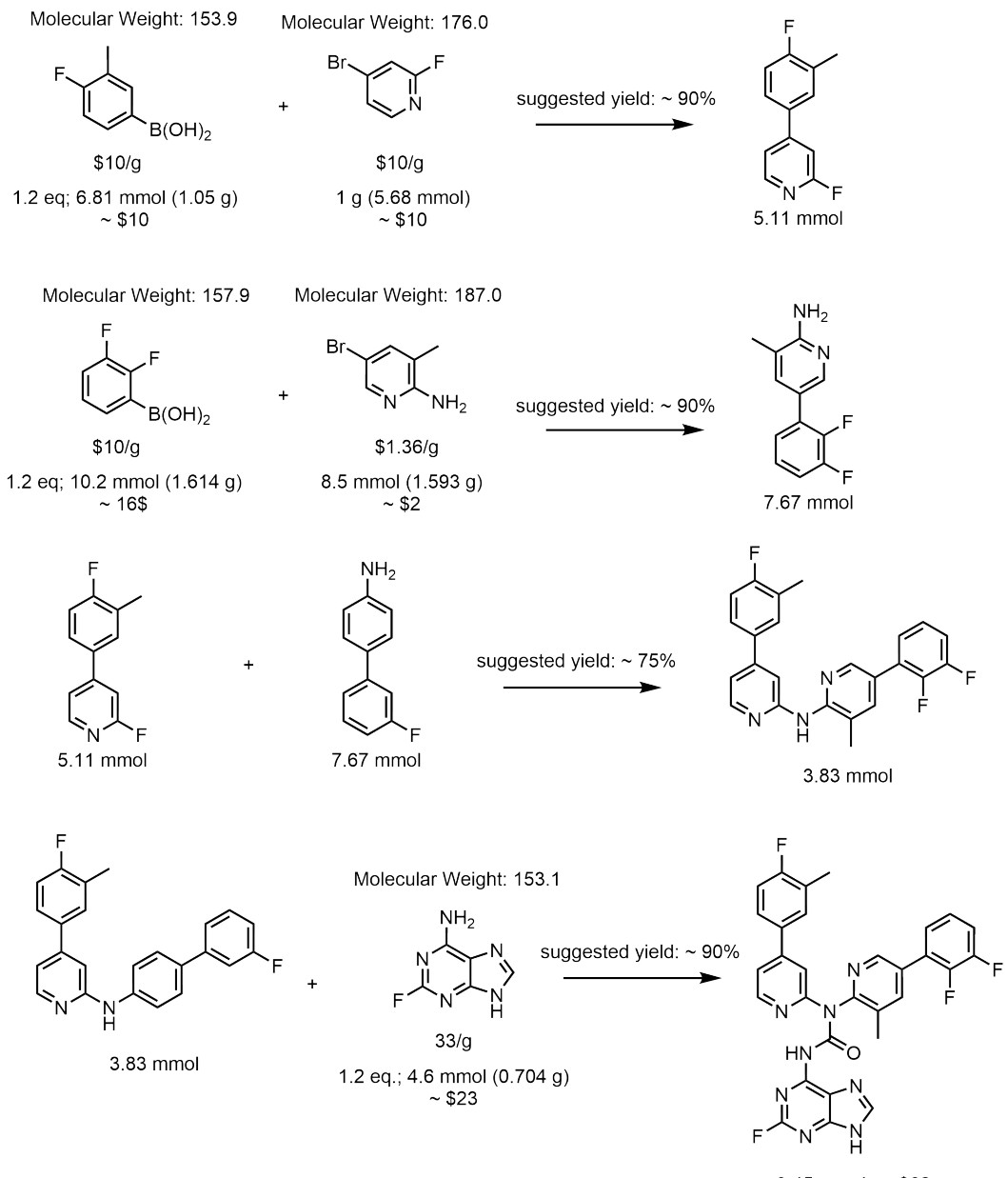

Figure 25: Plausible synthesis plan and estimated precursor cost for RGFN-produced ClpP ligand 5.

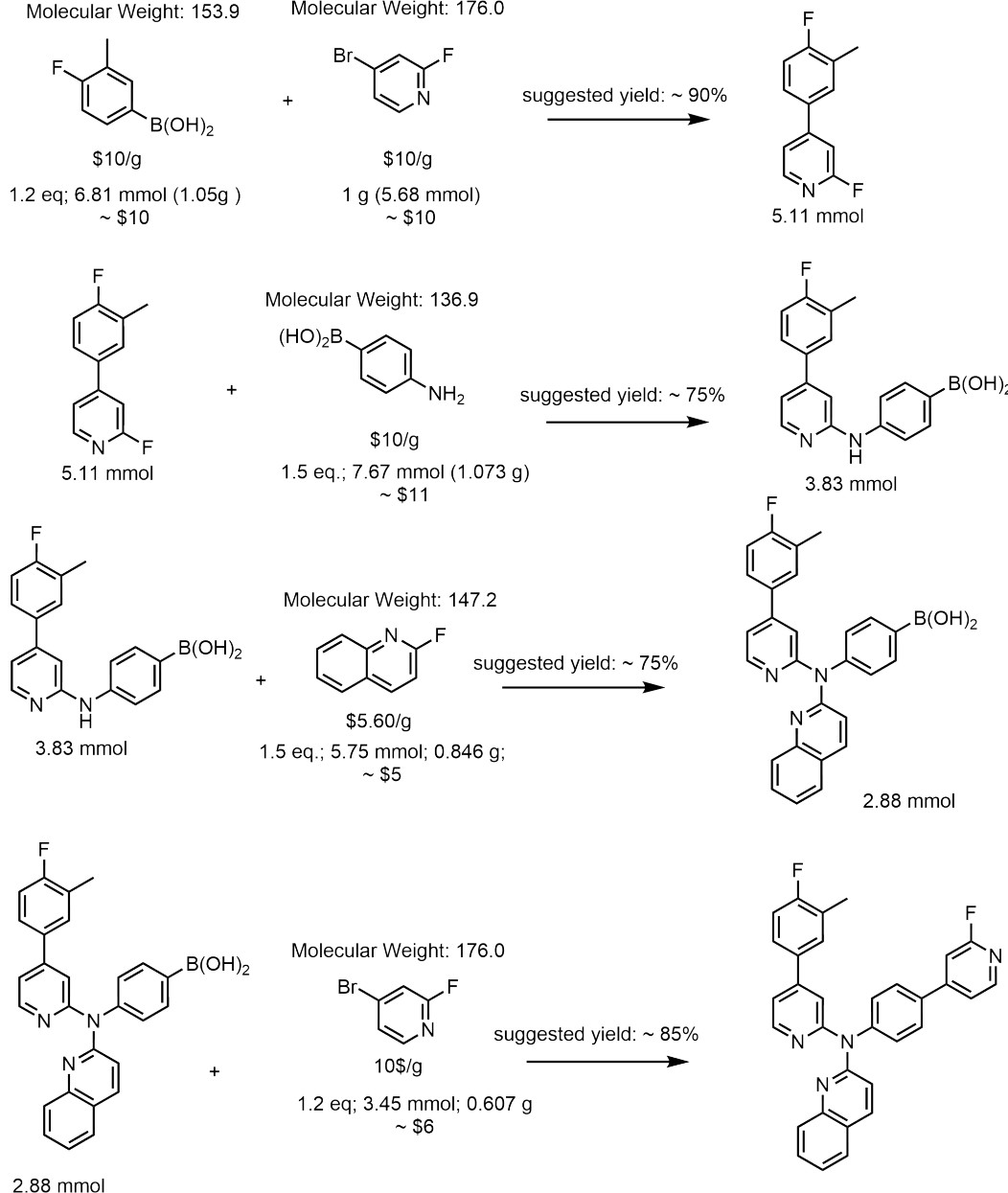

Figure 26: Plausible synthesis plan and estimated precursor cost for RGFN-produced ClpP ligand 6.

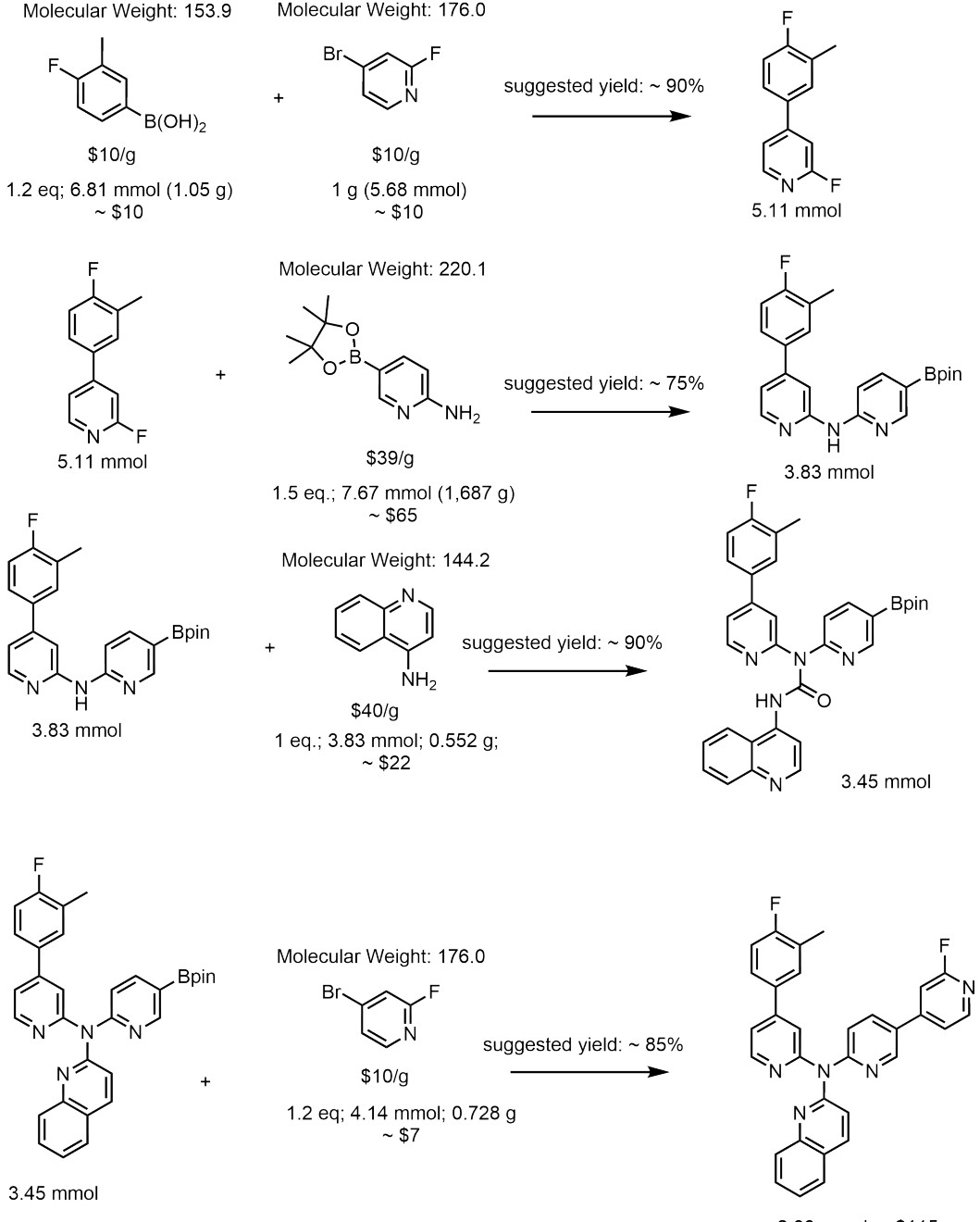

Figure 27: Plausible synthesis plan and estimated precursor cost for RGFN-produced ClpP ligand 7.

Molecular Weight: 157.9        Molecular Weight: 176.0

$10/g                          $10/g
1.2 eq; 6.81 mmol (1.076 g)    1 g (5.68 mmol)
        ~ $11                        ~ $10

suggested yield: ~ 90%

5.11 mmol

Molecular Weight: 136.9

5.11 mmol                      $10/g
                               1.5 eq.; 7.67 mmol (1.073 g)
                                        ~ $11

suggested yield: ~ 75%

3.83 mmol

Molecular Weight: 152.6

3.83 mmol                      $10/g
                               1.2 eq.; 4.6 mmol (0.702 g)
                                        ~ $7

suggested yield: ~ 75%

2.88 mmol

Molecular Weight: 176.0

2.88 mmol                      $10/g
                               1.2 eq; 3.45 mmol (0.607 g)
                                        ~ $6

suggested yield: ~ 85%

2.44 mmol, ~ $44

Figure 28: Plausible synthesis plan and estimated precursor cost for RGFN-produced ClpP ligand 8.

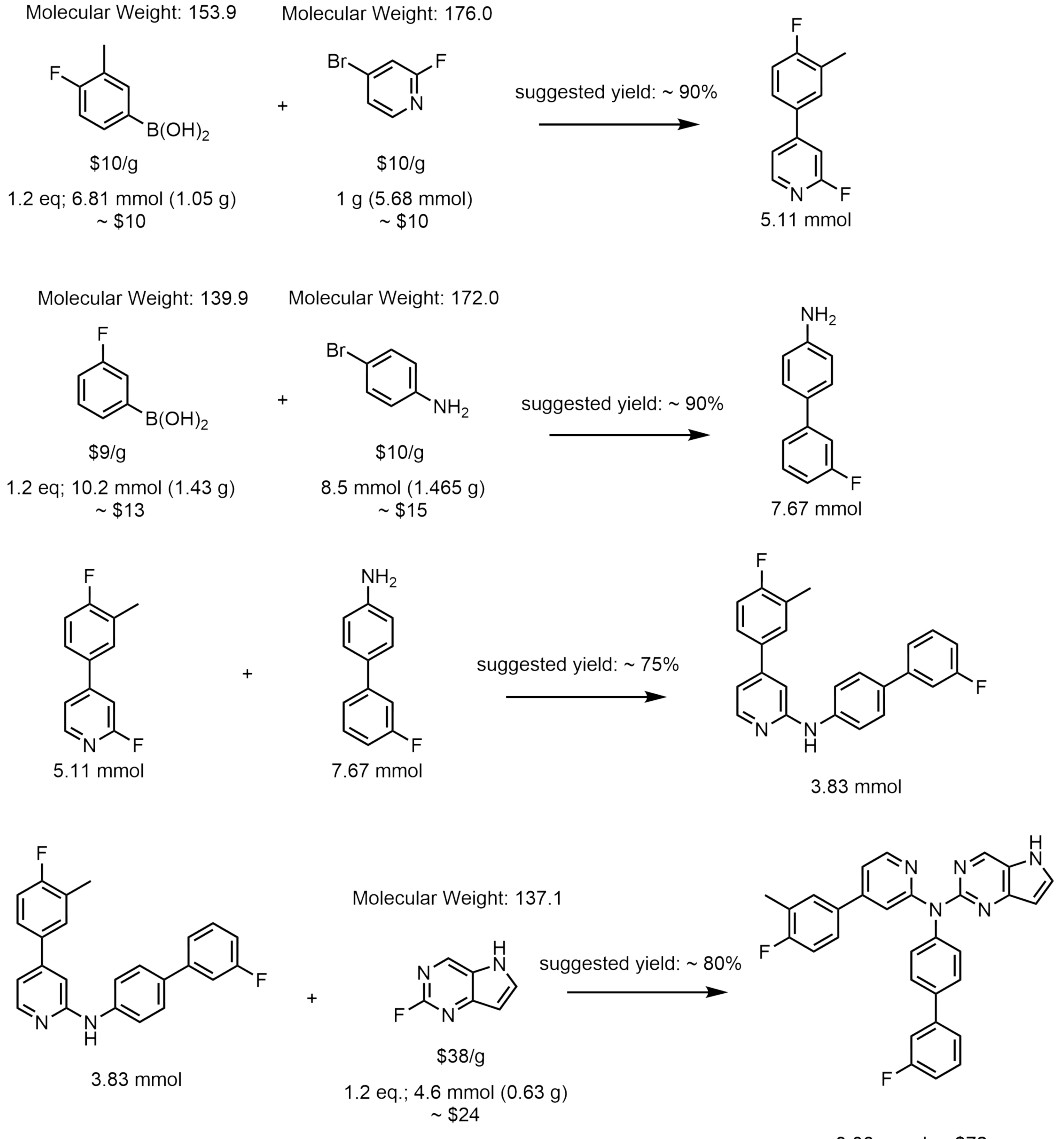

Figure 29: Plausible synthesis plan and estimated precursor cost for RGFN-produced ClpP ligand 9.

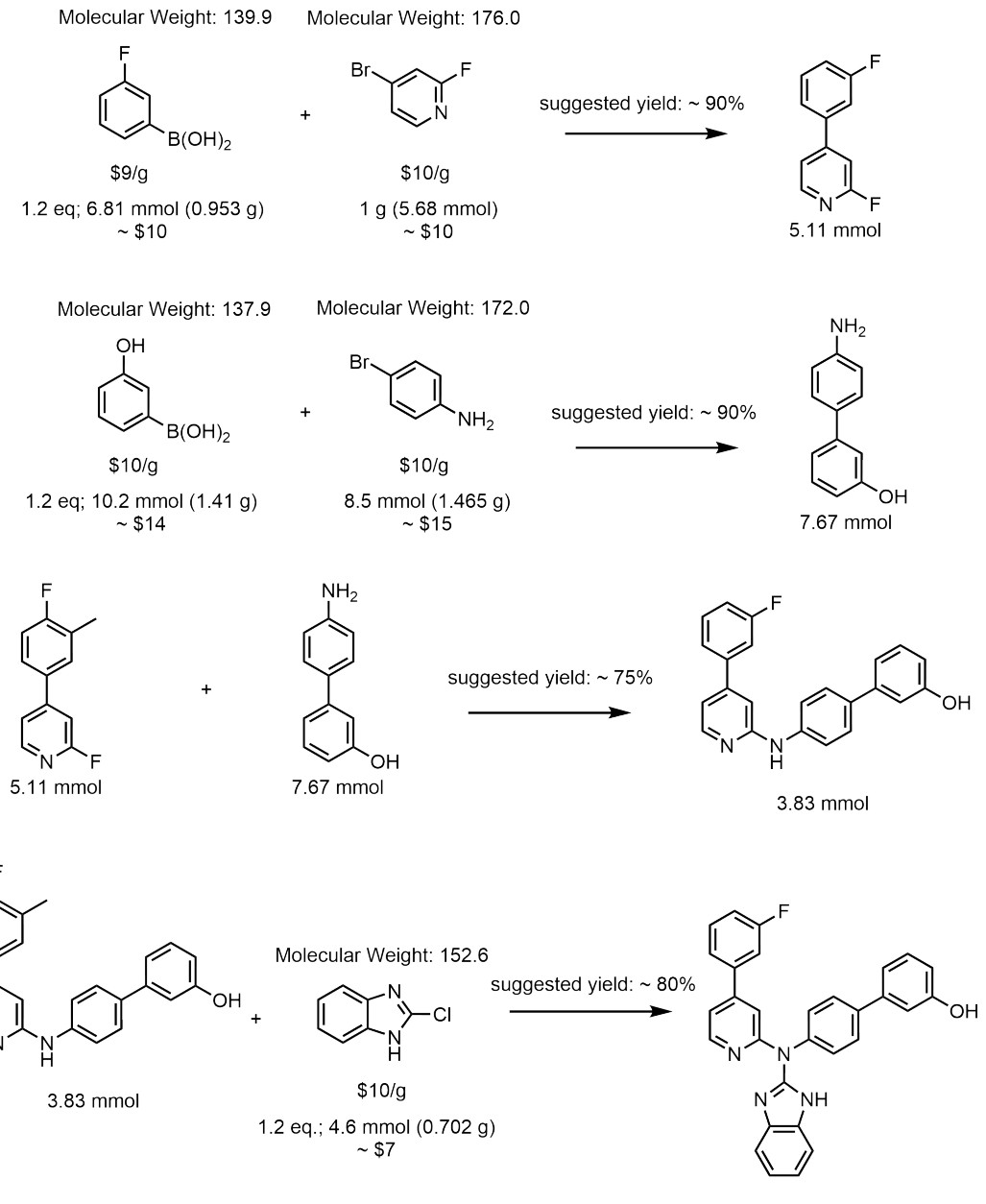

Figure 30: Plausible synthesis plan and estimated precursor cost for RGFN-produced ClpP ligand 10.

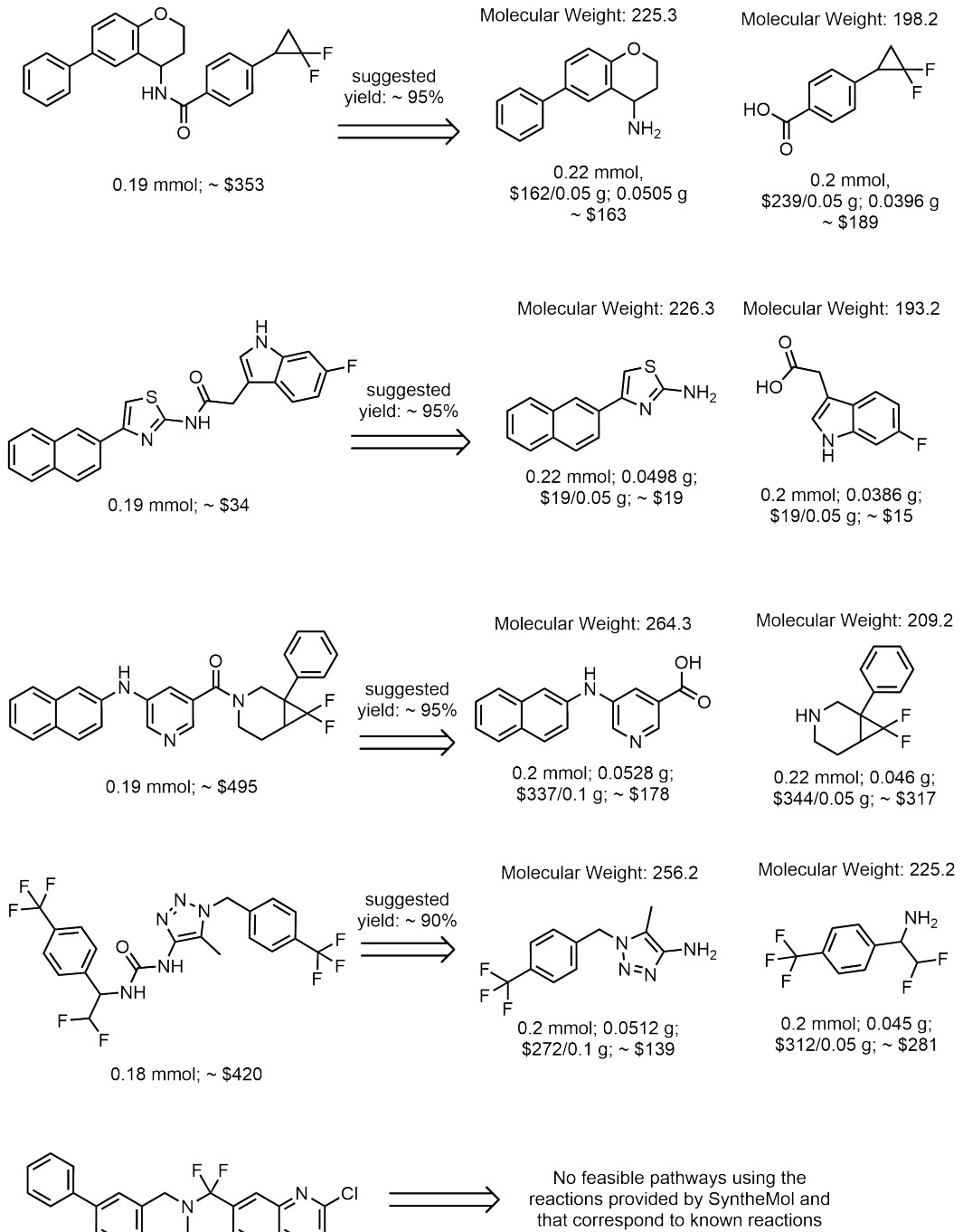

Figure 31: Plausible retrosynthesis plan and estimated precursor cost for SyntheMol-produced ClpP ligands 1-5.

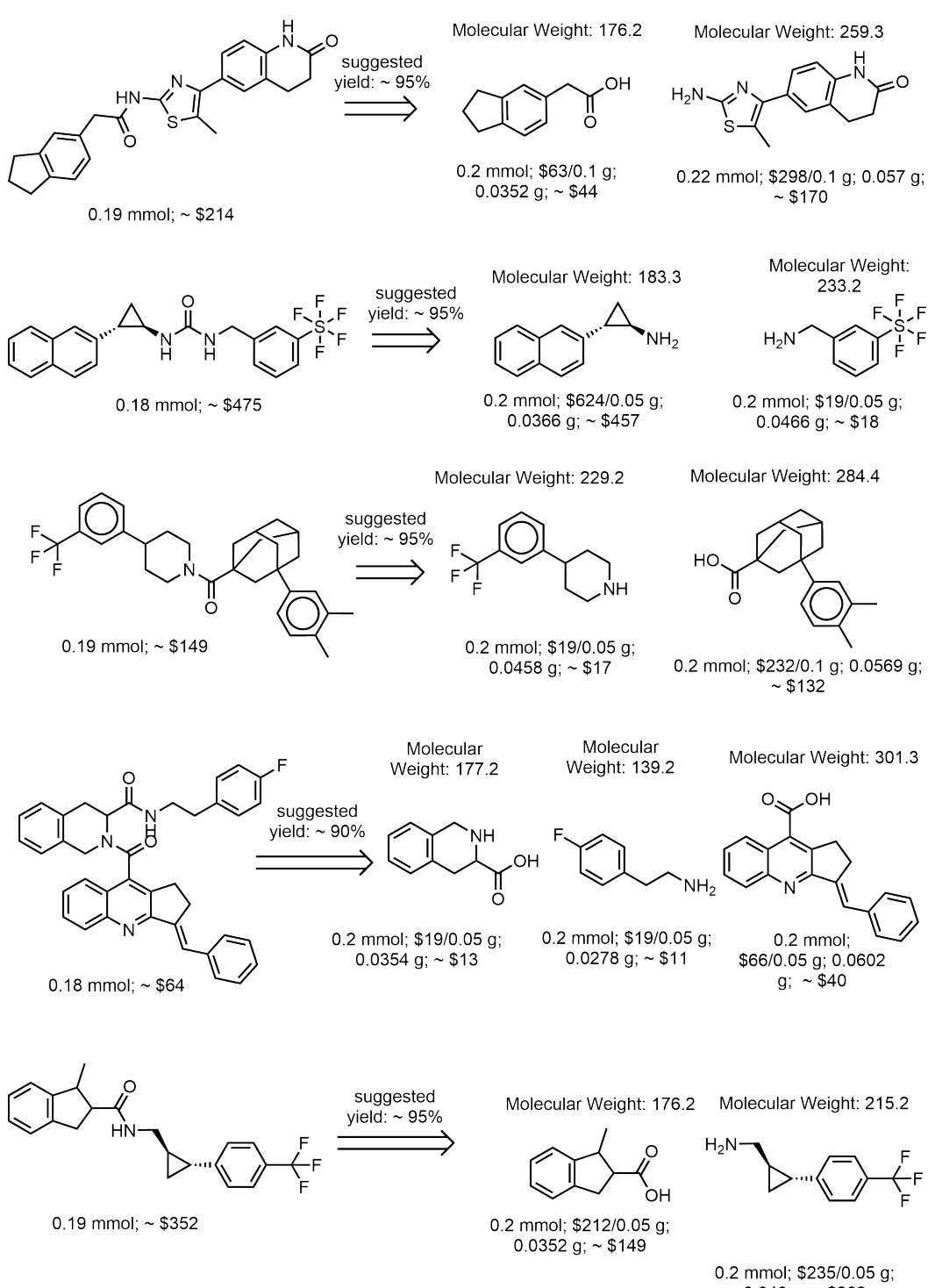

Figure 32: Plausible retrosynthesis plan and estimated precursor cost for SyntheMol-produced ClpP ligands 6-10.

