# OpenReview forum: "RGFN: Synthesizable Molecular Generation Using GFlowNets"
_NeurIPS.cc/2024/Conference — NeurIPS 2024 poster_

### Official Review · Reviewer_sx7b · 2024-06-28

**Soundness:** 3
**Presentation:** 2
**Contribution:** 3
**Rating:** 6
**Confidence:** 3

**Summary:**

This study modifies prior GFlowNet-based frameworks to produce molecules satisfying synthesizability. While previous GFlowNet-based molecular generation completes molecules by adding fragments or atoms, the proposed method repeats following stems to generate molecules: (1) selects reaction templates, (2) selects reactants, and (3) performs reactions and selects one of the resulting molecules. In the experiments, the proposed method shows competitive performance while preserving synthesizability.

**Strengths:**

- The proposed method first incorporates a reaction-based generation framework with GFlowNet for synthesizability.
- A chemical language for synthesizability seems to offer cost advantages in real-world applications, demonstrating that it acts as a promising inductive bias to generate promising molecules.
- The proposed method shows competitive performance compared to (1) FGFN in terms of synthesizability and (2) SyntheMol in terms of average reward and mode discovery. Although it finds fewer modes compared to FGFN, this seems to stem from constraints in the generation space.

**Weaknesses:**

- The preliminary is insufficient for understanding the components of the method, particularly mentioning the flow matching condition while the method section actually describes forward and backward policy.
- To improve synthesizability-related metrics, one may consider including them in the rewards. Can authors provide a comparison with FGFN including the SA score in the reward? (e.g., multiobjective GFN)
- The lower number of modes in RGFN (compared to FGFN) is due to constraints in the generation space (ensure high SA scores). Therefore, it would be interesting to compare the number of modes filtered by SA scores.
- Can authors compare the cost of generation time? Additionally, I am curious about time costs for (finding a synthesizability path + generation with FGFN) vs. (generation with RGFN).

**Minor:**
- Error in line 114 $m'i$

**Questions:**

See weaknesses.

**Limitations:**

The authors have acknowledged the limitations of their approach.

---

> ### Author Rebuttal · Authors · 2024-08-07
>
> # Weakness 1 Response:
> Thank you for this comment, we agree that the preliminary section present in the original manuscript was insufficient for understanding the method. We revised it to include the following information, which hopefully will address Reviewer’s remark:
> > Another way to rephrase the flow-matching constraints is to learn a forward policy $P_F(s_{i+1}|s_i)$ such that trajectories starting at $s_0$ and taking actions sampled by $P_F$ terminate at $x \in \mathcal{X}$ proportional to the reward.
>
> > Trajectory balance. Several training losses have been explored to train GFlowNets. Among these, trajectory balance [citation] has been shown to improve credit assignment. In addition to learning a forward policy $P_F$, we also learn a backward policy $P_B$ and a scalar $Z_\theta$, such that, for every trajectory $\tau = (s_0 \rightarrow s_1 \rightarrow \dots \rightarrow s_n = x)$, they satisfy:
> \begin{equation}
> Z_\theta \prod_{t=1}^{n} P_F(s_t|s_{t-1}) = R(x) \prod_{t=1}^{n} P_B(s_{t-1}|s_t)
> \end{equation}
>
> However, please note that we are still fairly constrained in terms of space, hence the description has to remain brief.
>
> # Weakness 2 Response:
> Thank you for the suggestion. We included a fragment-based GFN that used the average of proxy value and SA score as the reward as another baseline (specifically, we used $R(x) = exp(\beta * (0.5 * proxy(x) / max_{proxy} + 0.5 * (10 - SA_{score}) / 10)$, with $\beta$ adjusted per proxy to match the original reward range. We attached the results in the PDF with the main answer to the reviews. As can be seen, using the SA score as a part of the reward slightly reduces the obtained proxy values, while improving the SA scores (which is expected). However, as can be seen, it does not translate to AiZynthFinder scores on par with RGFN, which highlights the issue of poor reliability of SA scores that was discussed in the manuscript.
>
> # Weakness 3 Response:
> Thank you for the suggestion. As requested, we investigated computing the modes based on SA score-thresholded molecules only (with SA <3, which should correspond to easy-to-medium synthesis difficulty; lower values yielded similar trends). We conducted this experiment specifically for ClpP, with the results attached. However, this did not turn out to change the results significantly, since as discussed in the original manuscript, SA scores are a poor approximation of synthesizability. We could not conduct a similar experiment with AiZynthFinder in time due to much larger computational cost. We agree with the Reviewer that selecting only synthesizable modes should significantly improve RGFN’s performance, but the issue is once again evaluating the synthesizability.
>
> # Weakness 4 Response:
> On a single RTX Quadro 8000 GPU, generating 1000 molecules from a trained FGFN model took ~7 seconds, while evaluating the top 100 molecules with AiZynthFinder took 2206 seconds. On the other hand, generating 1000 samples using RGFN took ~14 seconds. It’s worth noting that FGFN used a heavily optimized implementation [1], while our approach didn’t leverage multiprocessing for sampling, which might have produced additional overhead. Still, this illustrates that when taking into account the cost of synthesis, using RGFN becomes much more efficient.
>
> # Minor
>
> Thank you for spotting this, it was revised!
>
> [1] https://github.com/recursionpharma/gflownet

---

> ### Comment · Reviewer_sx7b · 2024-08-08
>
> Thank you for the detailed and informative clarification. Most of my concerns are resolved. I believe that the paper has a basic value to be accepted. Therefore, I'd like to increase my score (5->6).

---

### Official Review · Reviewer_5soV · 2024-07-05

**Soundness:** 2
**Presentation:** 3
**Contribution:** 2
**Rating:** 6
**Confidence:** 4

**Summary:**

This paper presents a model for molecule design called Reaction-GFlowNet (RGFN). RGFN is an extension of the GFlowNet framework [4] (which has previously been used to generate molecules by building them out of small fragments) to generate molecules through virtual chemical reactions, ensuring that the molecules RGFN generates are more likely to be synthesizable. The paper evaluates RGFN against GraphGA (a graph genetic algorithm), FGFN (the original fragment-based GFlowNet), and SyntheMol (a recent synthesis-based generative model of molecules proposed in [54] based around Monte Carlo tree search). The paper shows RGFN finds competitively scoring molecules on two proxy* and one docking task (as expected the unconstrained GraphGA does best), while also ensuring the molecules it suggests do better on important synthesizability metrics.

* proxy meaning the task is to optimize against a graph neural network (GNN) oracle trained on a small amount of target data.

(Edited Aug 14 to update the score: see comments below).

**Strengths:**

## Summary of review
This paper addresses one of the limitations of GFlowNets by explicitly incorporating synthesis plans into the molecular generative process. However, this seems to be a fairly straightforward extension of the GFlowNets framework with synthesis-based generative models of molecules proposed elsewhere. The experiments in this paper compare the proposed method against only one of these existing models.

## Originality/Significance
RFGN is an extension of GFlowNets [4] to build molecules through chemical reactions rather than through combining small molecular fragments. This addresses a key limitation with the original GFlowNet, which the experiments show often generates molecules for which no synthetic route can be found.

Having said that, there have been many synthesis-based generative models of molecules proposed (e.g., [8,17,54,19,22] and others not cited -- see below), and RGFN seems a fairly straightforward combination of these ideas with GFlowNets.

> Korovina, K., Xu, S., Kandasamy, K., Neiswanger, W., Poczos, B., Schneider, J. and Xing, E. (2020) ‘ChemBO: Bayesian Optimization of Small Organic Molecules with Synthesizable Recommendations’, in International Conference on Artificial Intelligence and Statistics. AISTATS 2020

> Nguyen, D.H. and Tsuda, K. (2022) ‘Generating reaction trees with cascaded variational autoencoders’, The Journal of chemical physics, 156(4), p. 044117.

> Vinkers, H.M., de Jonge, M.R., Daeyaert, F.F.D., Heeres, J., Koymans, L.M.H., van Lenthe, J.H., Lewi, P.J., Timmerman, H., Van Aken, K. and Janssen, P.A.J. (2003) ‘SYNOPSIS: SYNthesize and OPtimize System in Silico’, Journal of medicinal chemistry, 46(13), pp. 2765–2773.


> Button, A., Merk, D., Hiss, J.A. and Schneider, G. (2019) ‘Automated de novo molecular design by hybrid machine intelligence and rule-driven chemical synthesis’, Nature Machine Intelligence, 1(7), pp. 307–315.
​

## Quality
The optimization tasks done to evaluate RGFN seem interesting and it's nice to see the molecules suggested also evaluated in terms of their diversity and synthesizability (using a retrosynthesis planner). While I think it is great that the tasks were more complicated than some of the simple ones often used elsewhere, it might have been nice to include some more commonly-used (yet still challenging) benchmarks too (such as the Therapeutics Data Commons benchmark -- citation below). This would make comparisons with existing models easier.

> Kexin Huang, Tianfan Fu, Wenhao Gao, Yue Zhao, Yusuf Roohani, Jure Leskovec, Connor W Coley, Cao Xiao, Jimeng Sun, and Marinka Zitnik.   Therapeutics data commons:  machine learning datasets and tasks for therapeutics. arXiv preprint arXiv:2102.09548, 2021.


## Clarity
Overall I thought the paper was well-written and easy to follow. In particular, I thought Section 3.2 was helpful in giving an overview of the approach (along with the informative Figure 1), and Equations 2–7 were helpful in explaining the parameterizations of the various networks involved. The experiments also seemed to be clearly presented. Readers unfamiliar with GFlowNets [4] would likely have to read about them elsewhere (lines 70–82 provide a short, although limited introduction). However, this seems reasonable due to the overall space available.

**Weaknesses:**

## Originality
As mentioned under the originality/significance heading in the section above, RGFN seems a fairly straightforward combination of the ideas of GFlowNets with recent synthesis-based generative models of molecules. It would have been helpful if there had been more focus on why synthesis-based GFlowNets were a better approach to take than the currently proposed alternatives to better promote the paper's significance. (Hence why I have gone with a low contribution score).

## Only one other synthesis-based generative model of molecules is compared against
The only other synthesis-based generative model for molecules that is compared against in the experiments is SyntheMol, and this baseline has been limited here in the amount of compute it uses. I can understand no comparison is done against [22,19] due to the lack of open-sourced code, but it would have been nice to have had a comparison against some of the other ones cited, e.g., [17] with its open sourced code available on GitHub (https://github.com/wenhao-gao/SynNet). It would also be nice to have a comparison to the other mentioned methods for encouraging synthesizability, e.g. via using scoring methods [34].

(This along with the fact that no code is available -- see below -- is why I have gone with a lower soundness score).

## No code is shared
As far as I'm aware there is no code currently available for RGFN? It does not seem to be provided as part of the submission (despite this box being ticked in the paper checklist)? I would find it impossible to reproduce the results of the paper using the details provided in Appendices B-D alone (e.g., I'm unsure of what building blocks and reaction templates to use or even how the molecules fed into the GNNs were featurized). One of the proxies was initially trained in an unsupervised manner on the ZINC dataset (Appendix B), but there are scant details on how this was done.

On a similar note, the compute resources used to train the models also seems to be missing. This is again marked as included in the paper checklist, so would appreciate being pointed towards this if I have made a mistake and missed this?


## Method is currently limited to a restricted synthetic space
RGFN uses only 350 building blocks and 17 reactions. This is low compared to the ~150k building blocks and 49 reactions used in [19], or the 5000 building blocks and 90 reactions used in [22]. The experiments done in Section 4.3 suggests the RGFN scales badly to larger building block libraries even with architecture modifications and larger reaction libraries do not seem to have been tried. This, combined with the fact that RFGN can only explore linear synthetic trees (i.e., multiple complex intermediates cannot be created and then combined) and bimolecular reactions restricts the molecules that the approach could explore. This might be an issue in more challenging optimization tasks.

**Questions:**

1. What is the distribution of the number of reaction steps typically used when finding useful molecules?



2. I don't completely follow how the size of the state space was calculated in Figure 2. Perhaps a cartoon would be useful in Appendix A. Is the state space size different to the total number of unique molecules that can be generated?



3. I'm confused as to why RGFN seems to often outperform FGFN in the docking/proxy scores of the molecules it finds. I would have thought the latter model is more flexible and so should be able to find better scoring molecules, even if they did not end up being synthesizable. The paper suggests that the library of fragments available for RGFN might be less suitable than those found in the building blocks picked for RGFN. Have different fragments been tried or other experiments carried out to investigate this hypothesis?



4. RGFN seems to produce molecules with lower QED scores than SyntheMol and FGFN: is there any intuition for why this might be the case?



5. Line 243 says:
> "All RGFN modes were additionally inspected manually by a chemist and confirmed as synthesizable, which indicates that AiZynth scores are likely underestimated."

Does this mean that a chemist took the proposed molecule and synthesized it in the lab? Were the proposed synthetic routes from RGFN used or were alternatives derived?



6. Line 282 states:
>  "It is also important to recognize that RGFN does not explicitly generate synthetic routes to the molecules."

I did not understand what was meant by this. I thought the whole point of RGFN is that it did explicitly generate synthetic routes to the final molecule through the sampled trajectory?



7. Equation 4 versus 7:
a. My understanding is that the architecture expressed in Equation 4 is used instead of that of Equation 7 (apart from in Section 4.3)?
b. The architecture expressed in Equation 7 seems to do better in Figure 5 so why not always use this?
c. Could also a similar principle be used to scale up the number of reaction templates available?



8. Line 221 discusses a procedure for picking molecular modes from the list of proposed molecules. Given modes have to be a certain Tanimoto distance apart, I assume this procedure is run greedily and is somewhat dependent on how the molecules are ordered?

**Limitations:**

Two limitations are discussed in Section 5: (i) that the model is currently limited to a very small reaction and building block library, and (ii) the issues inherent with docking oracles. Regarding the first of these, the authors discuss and evaluate a modification of the architecture to try to enable the method to scale (see Section 4.3), but performance still degrades as the initial fragment library increases and so this seems to remain an outstanding limitation (see also the weaknesses section above). The second limitation (docking oracles are not perfect) is not limited to RGFN and is a problem among similar models more generally. The authors discuss incorporating additional, more-accurate oracles in a multi-fidelity framework to resolve this which seems reasonable.

Another limitation which is not discussed is the sample efficiency of the proposed approach (which is again also a limitation for the baselines). The number of molecules the non-GFlowNet baselines visit during optimization for the different tasks is described in Section 4. These are high and range from 70k-400k, and I would be interested in how this compares to the two GFlowNet models.

---

> ### Author Rebuttal · Authors · 2024-08-07
>
> # Significance
> We thank the Reviewer for pointing out additional references that were included in the revised version. While the idea of extending GFNs to operate in the reaction space might seem straightforward, significant work was needed to implement the approach, not emphasized enough in the original manuscript. These include:
> - Handling multiple possible products of a given reaction, which results in a de facto non-deterministic environment. This is not an issue in FGFN, as operating on graphs allows us to specify the place of inserting a new fragment. We deal with this by introducing a product selection step (Eq. 5).
> - Since GFNs require computation of parent states and masking invalid actions, this required implementing efficient recursive decomposition of molecules into BBs and crafting specific SMARTS templates for reactions that would limit the possible number of parents.
> - Crafting a specific set of high-yield reactions and low-cost fragments that would enable synthesis.
> - Improving scalability by introducing fingerprint-based action embeddings for fragment selection.
>
> Additional discussion can be found in Overall Response 2.
> # Quality
> We included the DRD2 benchmark from TDC. We attach the results. The trends are comparable to other oracles.
> # W1
> Please refer to the answer above regarding the originality of the approach. The comment regarding needing to emphasize more the differences between our and existing methods, was addressed in the Overall Answer.
> # W2
> When implementing SynNet, we were unable to reproduce a decoder that reliably generated valid SMILES strings. As the initial trained weights provided did not match the most recent version of the US Enamine BB stock and generated nonsensical SMILES, we retrained the model’s four MLPs on the recent stock, which still did not result in reliable SMILES outputs. Due to time constraints, we instead opt for RXNGenerator, a VAE model designed for generating synthetic trees. For a general response regarding other synthesizable molecule generation methods, see Overall Response 2.
> # W3
> We indicated in the paper checklist that the code would be provided upon acceptance and hope it wasn’t misleading to mark it as such in the checklist. We provide the code as is in the AC comment.
>
> We did mistakenly omit the computational resources in the original manuscript, we apologize for that. Evaluating fragment scaling took approximately 800 RTX4090 hours in total. Remaining experiments took roughly 24 hours on Quadro RTX 8000 for sEH, DRD2 and senolytic proxies, and roughly 72 hours on an A100 for docking-based proxies.
> # W4
> See Overall Response 1.
> # Q1
> Vast majority of the molecules were generated using 4 reactions (maximum allowed number). We applied this constraint because larger molecules tend to achieve higher docking scores, and GFNs generate samples with the probability proportional to the reward, leading to a higher likelihood of generating larger molecules. Achieving higher size diversity would be possible by penalizing larger molecules.
> # Q2
> The state space size is precisely the number of unique molecules that can be generated from the listed reactions and reactants. We have tried to explain this more clearly in the revised text.
> # Q3
> Unfortunately, we were unable to explore using different FGFN fragments in time for the rebuttal. Other factors we suspect were 1) larger state space size of FGFN (which makes convergence more difficult), and 2) implementation differences (we based FGFN on [1], while RGFN was implemented from scratch).
> # Q4
> Our hypothesis is that the QED differences stem mostly from different average molecular weights. While we tried to achieve comparable weight ranges for different methods, it’s still task dependent (Table 1). SyntheMol uses shallow synthesis trees which lead to smaller molecules. We note that QED could easily be improved upon by adding it directly as a reward term.
> # Q5
> Synthetic routes from RGFN were used. Costs of synthesis were estimated according to literature reaction yields, building block availability, and unit price. The molecules generated in the paper were not synthesized, but we are in the process of doing that for another batch.
> # Q6
> It is unlikely actual synthetic routes would fully follow those generated by RGFN due to the linear nature of RGFN compared to a retrosynthetic analysis that would aim to maximize the yield RGFN is not intended to optimize the synthetic route but rather to generate synthesizable compounds.
> # Q7
> - Yes, we use the architecture expressed in Eq. 4 rather than Eq. 7, which outperforms with larger fragment libraries. However, for the fragment library outside Section 4.3 which contains 350 molecules, both architectures perform on-pair, as illustrated in Figure 5. We opt for Eq. 4 only because it is simpler. We note this in the text.
> - The embedding scheme proposed in Eq. 7 can be used for reaction templates. We did experiment with using RxnRep [2] embeddings in the early stage of the project but observed only a negligible performance improvement.
> # Q8
> We employed the standard techniques of the Leader algorithm as implemented in RDKit. While in theory this process is dependent on the order of the molecules, in practice the molecules are always shuffled and the standard deviation for the number of modes across 10 shuffled runs is less than 0.5%. We now add a note to this in the manuscript.
> # L1
> Regarding limited library size, please refer to the Overall Response. We agree with the oracles being a limiting factor, but as the Reviewer states, this is not a unique problem to RGFN.
> # L2
> The GFlowNet models each visited 400k molecules, the budget set for baselines, with the exception of SyntheMol. However, our models were able to converge much earlier, and allowing RGFN/FGFN to run for the remaining time was meant to allow exploration. We attach convergence plots for RGFN to illustrate this.
>
> [1] github.com/recursionpharma/gflownet
> [2] doi.org/10.1039/d1sc06515g

---

> > ### Comment · Reviewer_5soV · 2024-08-08
> >
> > Thanks to the authors for their detailed rebuttal and responding to each of my points. Particularly appreciate the additional experiments provided to address some of my/other reviewers' concerns.
> >
> > I have one minor follow up question regarding Q6, the answer to which I do not think I fully understand (partly also due to the response to Reviewer rpKJ's Question 2). I realize that RGFN might not necessarily generate the _best_ synthetic route to a proposed molecule, but I still do not understand why it does not generate _an_ explicit synthetic route. Are these not provided just by tracing back the sampling process for a product back to the initial building blocks?
> >
> > Some general comments on the other points made:
> > * I appreciate that restricting to a smaller fragment library increases robustness (global response), although I am glad to see you recognise that the one used here is particularly small for many use cases and are taking steps to address this (and also support more complicated synthesis plans in future work).
> > * Glad you will mention the related work initially overlooked. Pushing back though slightly on the importance of the template-based approach for reaction prediction used here, many of these other works often seem fairly agnostic to the reaction oracles used.
> > * Thanks again for the new results (particularly with the quick turnaround), and also for discovering and fixing a mistake with the FGFN baseline. I still think it's a little weird that FGFN does not do particularly well on optimization (Q3), but thanks for the intuition regarding the fragment library/implementation provided in your rebuttal.
> >
> > Will begin discussing the paper with the other reviewers and (unless anything new comes up from this) increase my score to reflect you addressing several of my points.

---

> > > ### Author Response · Authors · 2024-08-10
> > >
> > > Thank you for your comment. We would also like to take this opportunity to thank you again for a very thorough review, as that didn’t fit in the official answer. It is very appreciated and definitely helped us improve the paper.
> > >
> > > Answering the follow up question regarding Q6:
> > >
> > > Let us preface by saying that we suspect some of the ambiguity comes from different common understanding of the term “synthesis route” as used in ML and chemistry communities. In the ML community, the synthesis route is a sequence of reactions transforming sets of reactants into products. According to this definition, RGFN framework can output a synthesis route (albeit not necessarily the optimal one). However, in the answers and the paper we refer to the chemistry-based understanding of the term, elaborated on below. We apologize for the lack of clarity, we tried to be precise in chemical terminology, but recognize that given the audience we should clarify this, which we will do in the revised paper.
> > >
> > > To elaborate on the additional components needed for actual wet lab synthesis (which we refer to when we discuss a “synthesis route”):
> > >
> > > An explicit route is not generated, but rather the likely building blocks necessary to construct a molecule (and the knowledge of the reactions encoded) will be readily identifiable. This does not mean that an explicit synthetic route has been generated. A synthesis is more than just the building blocks and reactions used.
> > >
> > > 1. Firstly, consider a molecule ABCD constructed from four fragments (building blocks). A synthesis of such a molecule might proceed as A + B -> **AB** + C -> **ABC** + D -> **ABCD**. An alternative linear route might start with a coupling of B + C as the first step. Another alternative might employ a convergent synthesis, i.e., A + B -> **AB**, then C + D -> **CD**, and then **AB** + **CD** -> **ABCD**. There are other possibilities. Even if the same four building blocks are used for each these are not the same syntheses, and in practice they would show different degrees of success. A trained synthetic chemist would be able to suggest the best order of couplings.
> > >
> > > 2. In some cases “protection group” strategies or the use of surrogate functional groups and reactions may be needed or be more efficient.
> > >
> > > 3. In some cases alternative strategies to the synthesis might still be preferred. For example, in the convergent 3 step synthesis of **ABCD** shown above. If the final coupling is somehow incompatible (or not optimal) it might be necessary to employ a synthesis such as C + E -> **CE** and then do the coupling **AB** + **CE** -> **ABCE** followed by **ABCE** -> **ABCD**. Although this adds an extra reaction, if it avoids some incompatibility (or inefficiency) between the component **D** and coupling used, then this would be an acceptable solution (indeed issues like this are common – more often than not!). Again a trained synthetic chemist would be able to easily suggest such strategies (or alternatively software packages that explicitly design syntheses).
> > >
> > > 4. Careful choice of reaction conditions, external reagents, catalysts, etc. are also essential aspects of any synthesis. These are not established with RGFN.
> > >
> > > We hope that this answers the question.
> > >
> > > Also, regarding another remark:
> > >
> > > > Pushing back though slightly on the importance of the template-based approach for reaction prediction used here, many of these other works often seem fairly agnostic to the reaction oracles used.
> > >
> > > Could we please ask for a clarification on whether the Reviewer refers to the discussion in the shared rebuttal, Originality / Similar Works paragraph, specifically the reliance on pre-trained neural networks to predict synthetic trees? As we are not entirely sure to which point does that comment refer to.

---

> > > > ### Comment · Reviewer_5soV · 2024-08-12
> > > > **Thanks for your reply: responding to your explanation and request for me to clarify my point**
> > > >
> > > > Thank you for your response and kind words!
> > > >
> > > > **Re Q6:** This is helpful thanks and answers my question!
> > > >
> > > >
> > > > **Re my remark:**
> > > > Happy to clarify this point. This was in response to the global "Author Rebuttal by Authors" posted above, namely under the section "Originality / Similar Works" and the statements:
> > > > > We note that the majority (UniRXN, RXNGenerator, SynNet, ChemBO) rely on pre-trained neural networks to predict synthetic trees and therefore require the curation of large datasets of druglike molecules and generating ground-truth synthesis routes. ... . Moreover, while template-free methods like UniRXN, ChemBO, and MoleculeChef are capable of accurately predicting the products of reactions given reactants, they do not specify exactly the reaction used at each step - valuable information for chemists to prioritize easy-to-synthesize molecules out of a large pool of generated candidates.
> > > >
> > > > The particular form of the reaction oracle does not seem to be a key component of many current synthesis-based de novo design algorithms. To use methods which initially ran with template-free reaction oracles, such as ChemBO and MoleculeChef, with a template-based reaction oracle (that specified exactly the reaction used at each step) would be a very straightforward switch.
> > > >
> > > > Moreover, I don't believe that the requirement to curate large datasets is a particularly strong argument either. Yes, a model like SynNet uses approximately 200k synthesis plans for pre-training its NNs, but these synthesis plans are generated artificially and so this step seems on par with the 400k synthetic routes generated by GFlowNet in the process of its optimization. Likewise, ​​ChemBO does not actually require pre-training on synthetic routes.
> > > >
> > > > Regarding the significance/originality, I was somewhat more convinced by the arguments you made in the rebuttal to my review above, suggesting work was needed to develop GFlowNets to operate in reaction space that I might have overlooked (bullet points: 1, 2, and 4, starting with "Handling multiple possible products...").

---

> > > > > ### Author Response · Authors · 2024-08-12
> > > > >
> > > > > Thank you for your clarification! To address your points: yes, we agree that some of the existing methods (including the ones you mentioned) could potentially be extended to use e.g. our predefined set of high-yield chemical reactions and low-cost fragments. We would argue that in some sense, this makes our contribution of proposing a curated set of reactions and fragments more impactful, as it can benefit development of other methods. Moreover, even if it might seem “straightforward”, it required significant effort and chemical expertise to craft - so in a sense, it’s perhaps a slightly more interdisciplinary paper than a typical NeurIPS submission (but we would like to think that it makes it potentially more useful to propagate within the ML community).
> > > > >
> > > > > Regarding the dataset argument, we agree to some extent, but would like to point out that even such a large synthetic dataset would not guarantee generalization out of distribution. The ability to enforce hard domain constraints (by masking invalid actions) is one of the strengths of our method that makes it useful in this context. Finally, it's worth mentioning that the authors of ChemBO paper recognize this limitation:
> > > > >
> > > > > > While outputs of reactions are well known for simple cases, it is impossible to predict outcomes with complex molecules, and in some cases, the outputs may not even be deterministic. Fortunately however, there have been several advances in computational chemistry to predict outcomes of chemical reactions, which can be used in place of the oracle. In our work we use Rexgen [26]. It should be emphasized that since such predictors are not perfect, so in practice, ChemBO could end up recommending unsynthesizable molecules and/or incorrect synthesis recipes
> > > > >
> > > > > Thank you once again for an insightful discussion!

---

> ### Comment · Reviewer_5soV · 2024-08-14
> **Thanks for discussion and further results**
>
> Thank you for following up. Yes, while I agree your curated set of reactions and fragments could be an interesting contribution, currently it is hard for me to judge given that they are not actually included in the submission (related to point W3). I believe (from the initial rebuttal) you have provided this to the AC though (note I cannot see the AC comment) so leave it with them.
>
> Anyway, I have edited my score to reflect many of my original points getting addressed and to reflect the additional experiments carried out (including the one recently posted), showing that the diversity of the produced molecules could be a reason to prefer the approach here over other previously proposed synthesis-based de novo design methods.
>
> Thanks also for the discussion!

---

### Official Review · Reviewer_rpKJ · 2024-07-12

**Soundness:** 3
**Presentation:** 3
**Contribution:** 3
**Rating:** 6
**Confidence:** 3

**Summary:**

This work proposes a GFlowNet-based framework for synthesizable molecule design, which comprises of building block selection stages and reaction selection stages. The GFlowNet-based model uses chemical oracle functions as reward and the goal is to generate synthesizable molecules with the desired property quantified by the scoring function.

**Strengths:**

- This work tackles the synthesizability challenge in molecular design, which is crucial to experimental validation but has been long overlooked by previous work on molecular generation.
- Molecular synthesis pathways have been represented by directed acyclic graphs, which fit well the structure of GFlowNet sampling process. Therefore, this work choosing GFlowNet as the framework is proper and well-motivated.
- The experimental study is comprehensive. It demonstrates that RGFN can produce synthesizable molecules with higher docking score. It also examines the diversity and docking structures of the generated molecules

**Weaknesses:**

- As acknowledged in the paper, the major limitation is scalability to a larger chemical space. It becomes much more difficult to discover hit scaffolds when the chemical space gets larger. In addition, the capability of this framework is limited by oracle functions.

**Questions:**

- Molecules generated by building blocks and reactions are guaranteed to be highly likely synthesizable. Is it still necessary to evaluate empirical synthetic accessibility scores of these molecules? as these scores are generally inaccurate and they cannot tell much when a molecule already has synthesizability guarantee.
- Can this framework be applied to bottom-up synthesis planning?

**Limitations:**

See Weaknesses

---

> ### Author Rebuttal · Authors · 2024-08-07
>
> # Weakness 1 Response:
> Regarding limited library size, please refer to the Overall Response. We agree with the oracles being a limiting factor, but as the Reviewer states, this is not a unique problem to RGFN.
> # Question 1 Response:
> The reviewer raises a good point. We agree that the SA scores are inaccurate, and in our opinion using them is not really necessary. However, we have tried as best as possible to demonstrate the synthesizability aspect in a rigorous way, and including the SA scores was one data point (in addition to the AiZynthFinder and manually showing the synthesis paths of top-10 molecules). Huge differences in SA scores can thus serve as a rough guideline. We also anticipated that not providing SA scores might lead to reviewers requesting the scores be included in our analysis, since the application of such scores is  fairly common. We acknowledge that demonstrating synthesizability is difficult in general, and more work into this problem  is required (but is outside the scope of this paper). We have added some text to highlight the caveats with current synthesizability assessment methods.
> # Question 2 Response:
> At this juncture the platform does not automatically recommend specific synthetic routes.  However, a trained synthetic chemist can readily propose reasonable synthetic routes for any proposed structure from RGFN, particularly given that the reaction templates used are well known. Synthetic routes may also be proposed by well-developed stand alone retrosynthetic software packages.

---

### Official Review · Reviewer_FKST · 2024-07-13

**Soundness:** 2
**Presentation:** 3
**Contribution:** 2
**Rating:** 4
**Confidence:** 4

**Summary:**

This work proposes a workflow to synthesize molecules with reaction templates starting from some building blocks. Through this pipeline, the synthesizability of the generated molecules can be improved based on some experimental evidence of molecular drug design tasks.

**Strengths:**

(1) This reviewer agrees with the importance of researching the synthesizability of generated molecules;

(2) The presentation of the proposed approach is very clear. This reviewer can quickly understand the major idea of the whole workflow;

(3) The idea of generating molecules with chemical reactions is very valuable since this reviewer also thinks the most important thing is generating synthesizable molecules.

**Weaknesses:**

(1) The related work discussion in this work is insufficient. This reviewer just searches a bit on the web and find three highly related articles, which are "Bridging the gap between chemical reaction pretraining and conditional molecule generation with a unified model", "A generative model for molecule generation based on chemical reaction trees" and "Generating molecules via chemical reactions". This reviewer believes other relevant articles are not covered in this article. A more comprehensive and systematic related work section is required;

(2) The proposed method has many limitations, including reliance on reaction templates and powerful pre-trained models. We state these limitations in the following section;

(3) The evaluation of the proposed approach is not comprehensive enough. It does not include the conventional molecular generation benchmarks like novelty, validity, and diversity.

**Questions:**

(1) This reviewer is concerned about the pre-trained oracle models. It seems that almost every step of the pipeline relies on a single pre-trained model. So how to derive these pre-trained models independently?

(2) This reviewer is concerned about the ability to generate truly novel molecules. It seems the proposed approach starts with a selected set of fragments and deduces products through reaction templates. However, since reaction templates are extracted from known reactions, it seems nearly impossible to generate purely novel molecules through this pipeline. How do the authors address and evaluate this issue?

**Limitations:**

(1) It seems the whole pipeline of RGFN heavily relies on many different pre-trained models. Each stage can affect the overall process drastically.

(2) The proposed approach is based on reaction templates. The reaction templates must be updated frequently to cover more types of reactions. However, the template-based method has poor generalization to unseen reactions (new combinations of reactants).

---

> ### Author Rebuttal · Authors · 2024-08-07
>
> # Weakness 1 Response:
> Thank you for pointing out the omission of the Qiang et al., 2023 manuscript (“Bridging the gap..”) and the work by Nguyen et al. (“A generative model … reaction trees”) , we added the references to the revised version. In the related work section we cited a review by Meyers et al., 2021 (reference #42 in the original manuscript) that covers many other relevant works. The manuscript “Generating molecules via chemical reactions" by Bradshaw et al., was actually cited by us (reference #8). The cited manuscript (“A Model to Search for Synthesizable Molecules”) refers to the NeurIPS version, while the version pointed out by the reviewer (“Generating molecules via chemical reactions”) refers to an earlier ICLR workshop version. We also include several references mentioned by another Reviewer.
> # Weakness 2 Response:
> Please refer to Question 1 Response for the answer to the stated limitation being reliance on pre-trained models, and Limitation 2 Response to the stated limitation being reliance on reaction templates.
>
> Reaction templates are necessary to avoid post facto evaluation of synthesizability for analogs generated solely based on physico-chemical parameters without regard to the realities of chemical synthesis. Synthetic intractability has in fact been a major issue with previous generative methods. The use of reaction templates solves this problem without imposing detrimental constraints on chemical diversity. Reaction templates are thus inherent to our approach as the only reliable means to avoid infeasible structures and/or to enforce focussing on robust chemical reactions (or those available at a preferred chemical supplier, like Enamine).
>
> We have added additional text to the manuscript to clarify this important point on the challenges of real-world synthetic chemistry. We thank the reviewer for highlighting this issue and hope the reviewer concurs with our arguments.
> # Weakness 3 Response:
> We apologize for the lack of clarity on this point. By the definition of our search space, all molecules generated are inherently valid (e.g., correct valence, reasonable size, etc.) because all compounds are generated by bona fide synthetic reactions. We also demonstrate diversity of generated compounds in Figure 4, which shows RGFN finds more modes (i.e., molecules dissimilar to each other) compared to GraphGA and SyntheMol. We note that because the GFN model is trained from scratch, all generated molecules are “new” because the model is not trained on any known molecules. They are also highly different from known ligands, as demonstrated in Appendix L. We also note that the current largest space of synthesizable molecules at Enamine (~40 billion) is not systematic in terms of reaction combinations and is more than an order of magnitude smaller than even the limited RGFN space we present here. We have revised the text of our manuscript to better reflect these points.
> # Question 1 Response:
> We apologize for a lack of clarity on this key issue. While some of the considered oracles are indeed pre-trained ML models (sEH proxy and senolytic proxy), we specifically include the GPU-accelerated docking as an alternative reward function to alleviate this issue. Our approach is agnostic to the reward function, and could even use something like a QED score as the reward. We emphasize that all of the considered baselines also rely on the exact same oracle models (including the pre-trained proxies). We believe that a strength of our approach is its adaptability to all possible reward functions.
> # Question 2 Response:
> Although having a limited set of known reactions in principle does restrict diversity, the space of synthesizable molecules is still vast and greatly exceeds all synthesized molecules to date. A huge space of compounds remains to be queried for biological activity based only on established chemical reactions, due to the massive number of related building blocks that can be plugged into any given reaction. Thus, there are approximately 219M substances known to be produced and characterized in the literature [1] and the evaluated size of produced chemical space in RGFN is already larger, despite using only a modest number of input reactions and building blocks. The total chemical space encompassed by these reactions and associated reactants has barely been explored.
> # Limitation 1 Response:
> As above, we emphasize  that RGFN does not heavily rely on pre-trained models. Please refer to Question 1 Response for the answer to this perceived limitation.
> # Limitation 2 Response:
> We would like to clarify the difference between reactions and reactants. The number of feasible reactions is limited, the number of reactants that can be joined together by these reactions is very large, and hence the number of combinations is extremely large. To give a simple example, naturally occurring peptides are generated by a single reaction type (amide coupling) with only 20 reactants (the L amino acids) yet can linearly generate tremendous chemical diversity. The reaction templates need not be updated as more reactants are added - the diversity of possible reactants that can be accommodated alone enables a combinatorial explosion. In practical terms, it is also possible to add any reaction to the dataset using the SMARTS language, and those that we present in the paper are known to be generally very robust and reliable. While this method obviously doesn't allow us to discover ligands that could be produced from yet unknown reactions, it is not an aim of this work to devise new synthetic reaction methods. On the contrary, our stated goal is to ensure "out-of-the-box" synthesizability by using selected reactions that are more likely to deliver the product and thus shorten the discovery process, while at the same time ensuring massive chemical diversity.
>
> [1] https://www.cas.org/cas-data/cas-registry

---

> > ### Comment · Reviewer_FKST · 2024-08-12
> > **Reply to Authors**
> >
> > Thank you for providing a detailed rebuttal. Below are our reviewer's responses:
> >
> > (1) The reviewer expresses concern regarding the omission of related works, which may impact the novelty and significance of the proposed research. Merely including the omitted literature as references is insufficient. Further writing improvements are necessary to elucidate the major contributions of the proposed method compared to previous methods, thereby justifying its significance.
> >
> > (2) From a machine learning algorithm perspective, the technical contribution of the proposed method appears somewhat limited. The algorithm generates molecules using reaction templates, an approach widely adopted in template-based retrosynthesis analysis and forward reaction prediction. The reviewer does not identify significant algorithmic differences between the proposed method and template-based reaction modeling (both forward and backward).
> >
> > (3) The evaluations appear somewhat confusing and not comprehensive enough. Table 1 suggests that the proposed RGFN method does not achieve state-of-the-art performance in most tasks, causing the reviewer to question the method's superiority. Moreover, since the title emphasizes "molecular generation," typical empirical comparisons should include molecular generation benchmarks to demonstrate the basic VUN (Validity, Uniqueness, Novelty) of the proposed method. Building upon these fundamentals, an additional reasonable metric for evaluating synthesizability should be incorporated. Although the reviewer genuinely believes that molecular generations based on chemical reactions can be more reliable and synthesizable, the current evaluations are insufficient to demonstrate RetroGFN's superiority.
> >
> > In conclusion, the reviewer has decided to maintain their rating, and we kindly defer the decision to the AC.

---

> > > ### Author Response · Authors · 2024-08-12
> > >
> > > We thank the Reviewer for answering our rebuttal.
> > >
> > > Regarding points 1 and 2, we again apologize for the omission of some mentioned papers and we are keen on improving the writing as per the Reviewer’s valuable suggestions. However, we believe that it does not fundamentally alter the presentation of our main novel contributions, which as discussed in the reply to the Reviewer 5soV are:
> > > - Handling multiple possible products of a given reaction, which results in a de facto non-deterministic environment. This is not an issue in FGFN, as operating on graphs allows us to specify the place of inserting a new fragment. We deal with this by introducing a product selection step (Eq. 5).
> > > - Since GFNs require computation of parent states and masking invalid actions, this required implementing efficient recursive decomposition of molecules into BBs and crafting specific SMARTS templates for reactions that would limit the possible number of parents.
> > > - Improving scalability by introducing fingerprint-based action embeddings for fragment selection.
> > > - Crafting a specific set of high-yield reactions and low-cost fragments that would enable fast and cheap synthesis experimentally, which has been a long-standing challenge in de novo small molecule discovery.
> > >
> > > Regarding point 3, we would like to emphasize that based on the (noisy) synthesizability metrics, RGFN does in fact achieve state-of-the-art performance in terms of synthesizability (and cost of synthesis, as illustrated in the Appendix), while preserving high performance in terms of mode discovery (higher than baseline methods other than FGFN). It’s this favorable balance that makes RGFN, in our opinion, an impactful approach, as easy synthesizability is the most important practical factor (unsynthesizable molecules with high property scores are not useful experimentally). As discussed in the paper, QED, molecular weight are included for completeness, but are secondary to AiZynthFinder scores, which in our opinions are the best approximation of synthesizability currently available. We will further clarify this in the updated manuscript with the Reviewer’s comments in mind.
> > >
> > > As discussed in the first answer, we would like to clarify that the metrics we utilize here are **chemically more rigorous than the typical VUN metrics**. VUN metrics are more suitable for methods such as 3D diffusion-based generative modeling where the model is trained on a dataset and inference can generate invalid molecules. All molecules generated by RGFN are valid by design (validity), RGFN is not trained on a dataset of molecules but on a reward function (i.e. novelty is not directly applicable, in the sense all of the molecules are “new”), and we use diversity which is a much stricter version of uniqueness. Typical graph-based generation methods, including most GFlowNet papers, use these more strict metrics as adopted here ([1-6]). We would argue that the ones used in our paper are more appropriate for illustrating a good trade-off between synthesizability and diversity of generated molecules.
> > >
> > > Regarding the comment that “additional reasonable metric for evaluating synthesizability should be incorporated”, we are eager to improve our manuscript and would kindly ask for the Reviewer to provide some specific suggestions for what other chemically rigorous metrics could be used. We were unable to identify realistic approaches to reliably do this - lack of good synthesizability evaluation metrics is, in fact, the main motivation behind RGFN. As pointed out by Reviewer rpKJ, showing the synthesizability of RGFN might not even be strictly necessary, as it guarantees it out-of-the-box by operating directly in the space of high-yield chemical reactions, and synthesizability metrics are inherently noisy.
> > >
> > > We hope you were convinced by our rebuttal to the other points in the first review (e.g. pre-trained models, usage of reaction templates). Please, let us know if you have any questions.
> > >
> > > Thank you once again for your detailed review and for considering our revisions. We are active in any further discussions and would really appreciate it if AC and SAC could take our explanations and clarification into consideration!
> > >
> > > [1] Zhang, Zaixi, et al. "Molecule generation for target protein binding with structural motifs." ICLR (2023).
> > > [2] Guo, Jeff, et al. "Link-INVENT: generative linker design with reinforcement learning." Digital Discovery (2023).
> > > [3] Zhu, Yiheng, et al. "Sample-efficient multi-objective molecular optimization with GFlowNets." NeurIPS (2024).
> > > [4] Korovina, Ksenia, et al. "ChemBO: Bayesian optimization of small organic molecules with synthesizable recommendations." PMLR (2020).
> > > [5] Nguyen, Dai Hai, and Koji Tsuda. "Generating reaction trees with cascaded variational autoencoders." The Journal of Chemical Physics (2022).
> > > [6] Swanson, Kyle, et al. "Generative AI for designing and validating easily synthesizable and structurally novel antibiotics." Nature Machine Intelligence (2024).

---

### Author Rebuttal · Authors · 2024-08-07

Thank you very much for your detailed feedback on our manuscript. We understand and have carefully considered your concerns with our work, and aim to provide a summarized response to the most common points here.
# Small Fragment Library
We agree with the Reviewers that the search space is indeed constrained as it stands. However, we wish to note that this constraint was purposeful to ensure practical fast synthesis for the generated molecules. That is, the end goal is to allow a typical chemical laboratory to quickly synthesize de novo molecules and test their properties experimentally. It is reasonable to expect such a laboratory to own hundreds of building blocks and conduct a handful of very reliable organic reactions, whereas it is impractical to employ >1000s building blocks and >20s reactions, as many building blocks/reactions require specialized synthetic expertise. We agree with the Reviewers that the trade-off is a limited search space, but our method nevertheless yields billions to trillions of synthesizable molecules, exceeding all current experimental techniques (e.g., DNA encoded libraries) which already yield strong drug candidate molecules for many targets. We do plan to incorporate non-linear synthetic trees and multi-component reactions but these aspects need careful experimental validation to ensure true synthesizability and robustness.
# Originality / Similar Works
We thank the Reviewers for highlighting additional relevant papers that we overlooked. We note that the majority (UniRXN, RXNGenerator, SynNet, ChemBO) rely on pre-trained neural networks to predict synthetic trees and therefore require the curation of large datasets of druglike molecules and generating ground-truth synthesis routes. This is in part an additional drawback of fragment and reaction datasets too large to serve as discrete action spaces. Moreover, while template-free methods like UniRXN, ChemBO, and MoleculeChef are capable of accurately predicting the products of reactions given reactants, they do not specify exactly the reaction used at each step - valuable information for chemists to prioritize easy-to-synthesize molecules out of a large pool of generated candidates. We present a simple alternative to these methods that avoids both the time cost of pre-training and provides template-based synthetic routes.
# Scalability to Larger Fragment Libraries
As stated in the manuscript, we recognize that one of the limitations of our approach is degrading performance with very large fragment libraries. However, as explained before, it is not necessarily our goal to support very large libraries in the first place, as they can be impractical if one wants to achieve rapid in-lab synthesis. Still, we recognize that their support can be useful in some settings, and provide a mechanism for improving scalability (using fingerprint-based action embeddings). We additionally would like to point out that 1) we believe decreasing performance is somehow expected, since by adding more fragments we increase the state space size exponentially, and it takes the model longer to converge, 2) still, even in the considered setting, the number of discovered modes remained large, suggesting the feasibility of using large libraries to increase chemical diversity (while preserving a high number of discovered modes). First point was illustrated in another experiment we conducted (attached with the figures), in which we count the number of discovered high-reward scaffolds in the last 10k samples throughout training. There we can see that as the model trains, the gap between small and large fragment sets decreases.
# Additional Results
As requested by the Reviewers, we included FGFN that used SA score as one of the reward terms as one of the baselines. We considered one more task, DRD2 oracle from TDC repository, as requested by another Reviewer (with results for SyntheMol still computing). We also included an additional baseline, RxnGenerator [2], but due to a very slow generation speed at this time only obtained results for the DRD2 task with a small number of molecules generated. While it achieves similar synthesizability, it discovers significantly less modes. Finally, we’d like to indicate that there was an error with the original results presented for FGFN in the senolytic generation task, in which we didn’t scale the proxy values properly. We attach the updated plots, but would like to point out that it did not change the overall conclusions, as FGFN still discovers only a very small number of high reward molecules. Still, we apologize for that mistake.

[1] Huang, Kexin, et al. "Therapeutics data commons: Machine learning datasets and tasks for drug discovery and development." arXiv preprint arXiv:2102.09548 (2021).
[2] Nguyen, Dai Hai, and Koji Tsuda. "A generative model for molecule generation based on chemical reaction trees." arXiv preprint arXiv:2106.03394 (2021).

---

### Author Response · Authors · 2024-08-14
**Requested additional experiments**

We wanted to provide updated results for the experiments requested by Reviewers (added DRD2 task, and two additional baselines: RxnGenerator [1] and FGFN with the SA score included as a reward term). By now, all of the runs except RxnGenerator in the ClpP task were completed. Crucially, similar to SyntheMol, due to very large computational overhead we capped the run time of RxnGenerator at approximately 24 hours per run (which is above the total RGFN run time). For the final paper, we intend to actually further increase the run time of RxnGenerator, but based on the current results (smaller number of molecules visited and significantly lower average rewards), we believe any changes in the conclusions to be very unlikely. Since we are not allowed to provide any figures at this point, we instead show a table with two most relevant metrics (AiZynthFinder scores and the number of discovered modes).

In general, the trends are comparable to the ones observed in the previous experiments: RGFN shows synthesizability scores comparable with other methods providing synthesizability guarantees (SyntheMol and RxnGenerator), illustrating the benefits of amortization of the GFlowNet framework. While in the sEH task AiZynth scores of RGFN are slightly lower, we suspect that it is mostly due to higher average weights of the generated molecules (since larger molecules are more difficult for AiZynthFinder to predict paths for). At the same time RGFN provides a significantly larger number of discovered modes (lower only than FGFN). Crucially, including SA score as a reward term for FGFN only marginally improves synthesizability (due to the unreliability of SA scores), at the same time decreasing the number of discovered modes. **Overall, this illustrates the usefulness of the RGFN framework, which has synthesizability guarantees similar to other synthesizability-focused methods, while producing significantly more diverse candidate molecules** (and does so using cheaper building blocks).

| Task  | Method       | Mol. weight $\downarrow$ | AiZynth $\uparrow$ | Modes $\uparrow$ |
|-------|--------------|--------------------------|--------------------|------------------|
| DRD2  | GraphGA      | 475.4 ± 53.2             | 0.41               | 664              |
|       | SyntheMol    | 365.6 ± 54.3             | 0.66               | 728              |
|       | RxnGenerator | 418.1 ± 53.3             | **0.87**           | 36               |
|       | FGFN         | 386.5 ± 45.0             | 0.76               | 11396            |
|       | FGFN+SA      | 381.1 ± 35.1             | 0.78               | 5903             |
|       | RGFN         | 447.1 ± 45.7             | **0.87**           | 6076             |
| sEH   | GraphGA      | 528.6 ± 42.3             | 0.04               | 1327             |
|       | SyntheMol    | 411.1 ± 66.7             | *0.80*             | 2336             |
|       | RxnGenerator | 421.6 ± 103.4            | **0.82**           | 5                |
|       | FGFN         | 473.4 ± 58.9             | 0.14               | 21548            |
|       | FGFN+SA      | 473.7 ± 62.2             | 0.27               | 5796             |
|       | RGFN         | 495.2 ± 49.6             | 0.56               | 2875             |
| Seno. | GraphGA      | 485.7 ± 75.6             | 0.05               | 347              |
|       | SyntheMol    | 441.4 ± 83.5             | 0.53               | 598              |
|       | RxnGenerator | 431.5 ± 100.9            | **0.65**           | 12               |
|       | FGFN         | 468.9 ± 47.7             | 0.02               | 44               |
|       | FGFN+SA      | 451.8 ± 54.5             | 0.13               | 14               |
|       | RGFN         | 558.7 ± 62.8             | *0.58*             | 2276             |
| ClpP  | GraphGA      | 521.0 ± 31.8             | 0.00               | 1571             |
|       | SyntheMol    | 458.2 ± 60.7             | *0.56*             | 1107             |
|       | RxnGenerator | -                        | -                  | -                |
|       | FGFN         | 548.6 ± 42.9             | 0.25               | 16960            |
|       | FGFN+SA      | 509.2 ± 52.4             | 0.33               | 11959            |
|       | RGFN         | 526.2 ± 37.6             | **0.65**           | 2880             |

[1] Nguyen, Dai Hai, and Koji Tsuda. "Generating reaction trees with cascaded variational autoencoders." The Journal of Chemical Physics 156.4 (2022).

---

### Decision · Program_Chairs · 2024-09-25

**Decision:**

Accept (poster)

**Comment:**

This paper studies generative models for molecular discovery, focusing on incorporating synthesizability by generating molecules within the chemical reaction space. The proposed method, RGFN, performs molecular generation explicitly in this space.

Given that this paper is on the borderline, we had a detailed AC-reviewer discussion to carefully assess pros and cons of the manuscript. The reviewers mainly identified the following concerns:
- **Evaluation on the AiZynth Score:** The evaluation is not entirely convincing. Specifically, it is important to clarify whether the reaction templates used in the GFlowNet framework overlap with those in AiZynthFinder and whether the baseline models are appropriately tuned, as these factors could significantly influence the scores.
- **Novelty:** The novelty of the approach is somewhat limited, as it is a relatively straightforward extension of the existing GFlowNet method.
- **Discussion of Related Work:** The initial submission lacked a sufficient discussion of related work, although this was adequately addressed in the authors' rebuttal.

In my opinion, these issues are not prohibitive for publication. For a new method, especially in empirical research, comprehensive evaluation is crucial, while it is not mandatory for the proposed method to outperform all existing approaches. Rather, it is essential that the method is reliable, which I believe is an important contribution from the ML side, and this paper satisfies it.

Considering these points, I recommend accepting the paper. Please ensure that the final version includes revisions based on the reviewers' suggestions and incorporates the additional empirical results provided in the authors' rebuttal, particularly those in the PDF.